# The tumor suppressor menin prevents effector CD8 T-cell dysfunction by targeting mTORC1-dependent metabolic activation

Junpei Suzuki[1,2,3], Takeshi Yamada[4], Kazuki Inoue[5], Shogo Nabe[1], Makoto Kuwahara[2,3,6], Nobuaki Takemori[7], Ayako Takemori[7], Seiji Matsuda[8], Makoto Kanoh[4], Yuuki Imai[5], Masaki Yasukawa [1] & Masakatsu Yamashita[2,3,6]

While menin plays an important role in preventing T-cell dysfunction, such as senescence and exhaustion, the regulatory mechanisms remain unclear. We found that menin prevents the induction of dysfunction in activated CD8 T cells by restricting the cellular metabolism. mTOR complex 1 (mTORC1) signaling, glycolysis, and glutaminolysis are augmented by menin deficiency. Rapamycin treatment prevents CD8 T-cell dysfunction in *menin*-deficient CD8 T cells. Limited glutamine availability also prevents CD8 T-cell dysfunction induced by *menin* deficiency, and its inhibitory effect is antagonized by α-ketoglutarate (α-KG), an intermediate metabolite of glutaminolysis. α-KG-dependent histone H3K27 demethylation seems to be involved in the dysfunction in *menin*-deficient CD8 T cells. We also found that α-KG activates mTORC1-dependent central carbon metabolism. These findings suggest that menin maintains the T-cell functions by limiting mTORC 1 activity and subsequent cellular metabolism.

[1] Department of Hematology, Clinical Immunology and Infectious Diseases, Graduate School of Medicine, Ehime University, Shitsukawa, Toon City, Ehime 791-0295, Japan. [2] Department of Immunology, Graduate School of Medicine, Ehime University, Shitsukawa, Toon City, Ehime 791-0295, Japan. [3] Department of Translational Immunology, Translational Research Center, Ehime University Hospital, Shitsukawa, Toon City, Ehime 791-0295, Japan. [4] Department of Infections and Host Defenses, Graduate School of Medicine, Ehime University, Shitsukawa, Toon City, Ehime 791-0295, Japan. [5] Division of Integrative Pathophysiology, Department of Proteo-Inovation, Proteo-Science Center, Ehime University, Toon City, Ehime 791-0295, Japan. [6] Division of Immune Regulation, Department of Proteo-Inovation, Proteo-Science Center, Ehime University, Toon City, Ehime 791-0295, Japan. [7] Division of Proteomics Research, Department of Proteo-Medicine, Proteo-Science Center, Ehime University, Toon City, Ehime 791-0295, Japan. [8] Department of Anatomy and Embryology, Graduate School of Medicine, Ehime University, Shitsukawa, Toon City, Ehime 791-0295, Japan. Correspondence and requests for materials should be addressed to M.Y. (email: yamamasa@m.ehime-u.ac.jp)

After antigen recognition, naive T cells initiate a cell intrinsic program that induces cellular expansion and differentiation into effector T cells. The activation and differentiation of T cells is regulated in a context-dependent manner. Environmental changes cause substantial alterations in the T-cell activation and differentiation process including dysfunction. Various states of T-cell dysfunction have been reported as a consequence of altered activation and differentiation processes, being characterized by terms such as anergy, tolerance, exhaustion, and senescence[1].

Aging-associated dysfunction in the immune system particularly affects the T-cell compartment and is involved in the age-related decline in the immune functions, which increase the susceptibility of elderly individuals to infectious diseases and certain cancers[2,3]. More recent reports on chronic viral-[4–6], chemotherapy-[7], or tumor-induced T-cell senescence[8,9] have suggested that T-cell senescence was also induced in an age-independent manner. A major characteristic feature of T-cell senescence is the acquisition of a senescence-associated secretory phenotype (SASP)[10], which is characterized by a striking increase in the secretion of pro-inflammatory cytokines, chemokines, matrix remodeling factors, and pro-angiogenic factors[11,12]. These factors deleteriously alter tissue homeostasis, leading to chronic inflammation and cancer[11,13–15]. Senescent T cells induce an increased susceptibility to autoimmune diseases such as rheumatoid arthritis through SASP[16–19]. Another major alteration in senescent T cells is the impaired IL-2 production and memory formation against infection. Therefore, the efficacy of vaccination is reduced due to senescence[5].

The serine threonine kinase mechanistic target or rapamycin (mTOR) is a key regulator of cellular metabolism[20,21]. mTOR signaling is required to integrate immune signals and metabolic cues for proper maintenance and activation of T cells[22–24]. mTOR exists in two multi-protein complexes: mTROC1 and mTORC2. Both mTORC1 and mTORC2 are activated within minutes of TCR stimulation. Recent studies have revealed that mTORC1 is a major regulator of aging and cellular senescence[25,26]. Rapamycin and other rapalogs specifically suppress the activity of the mammalian target of mTORC1 and decelerate cellular senescence[27,28]. Furthermore, mTORC1 inhibition is considered a viable strategy for preventing T-cell dysfunction and subsequent increases in the risk of age-related diseases and it was recently reported that mTOR inhibition improves the immune function in the elderly[29].

Emerging evidence suggests that T cells dramatically alter their metabolic activity during T-cell receptor (TCR)-mediated activation[30–32]. This change in the metabolic status is termed "metabolic reprogramming" and plays an important role in the regulation of T-cell-mediated immune responses[33]. Similar to other non-proliferating cells, naive T cells use fatty acid oxidization and/or a low rate of glycolysis and subsequently oxidize glucose-derived pyruvate via oxidative phosphorylation (OXPHOS) to generate ATP[34,35]. Upon activation, T cells immediately shift their metabolic program to anabolic growth and biomass accumulation, which support the rapid expansion of these cells and the acquisition of the effector function[34,36], known as the Warburg effect[37]. Although aerobic glycolysis is the dominant pathway of glucose metabolism in effector T cells, OXPHOS continues to occur[38,36]. Dysregulated T-cell metabolism is associated with impaired immunity both in cancer and chronic infection[33,39].

The nonessential amino acid glutamine is the most abundant amino acid in the blood and serves as a source of carbon and nitrogen for the synthesis of proteins, lipids, and amino acids[40]. Proliferating cells import extracellular glutamine and catabolize it via glutaminolysis in both the cytosol and mitochondria[41].

Glutaminolysis consists of two steps[42]. In the first step, glutamine is converted into glutamate, and in the second step, glutamate is catabolized into the tricarboxylic acid (TCA) cycle intermediate α-ketoglutarate (α-KG), which is consumed through OXPHOS or a reductive TCA cycle[42,43]. Activating and effector T cells also rapidly take up glutamine, and glutamine is required for maximizing the cell growth and proliferation of T cells[38,44]. In addition, α-KG regulates the enzymatic activity of Tet methylcytosine dioxygenase 2 (TET2), JumonjiC (JmjC) family histone demethylases, and PDH hydroxylases, suggesting that glutaminolysis is involved in the cellular differentiation processes[45].

Menin, a tumor suppressor, acts as a multifunctional scaffold protein and controls cell signaling and gene expression[46]. Certain germinal mutations of *MEN1*, which encodes MENIN, cause multiple endocrine neoplasia type 1[47], which is an autosomal dominant syndrome characterized by concurrent parathyroid adenomas, gastroenteropancreatic tumors, and several other tumor types. Menin interacts with H3K4 methyltransferases, including mixed-lineage leukemia 1(MLL1)[48], and is an oncogenic cofactor for MLL-associated leukemogenesis[49,50]. Menin is also known to be associated with the JunD proto-oncogene product (JUND), nuclear factor of kappa light poly peptide gene enhancer in B-cells 1 (NF-κB), peroxisome proliferator-activated receptor gamma (PPAR-γ), SMAD family member 3 (SMAD3) and β-catenin, indicating its involvement in transcriptional activation and repression[51,52]. It was also reported that menin localizes in the cell membrane compartment and inhibits Akt activation[53,54]. More recently, we reported a critical role of menin in regulating CD4 T-cell senescence[10].

In the present study, we examined the molecular mechanism by which menin inhibits the induction of CD8 T-cell dysfunction, including senescence and exhaustion with a focus on cellular metabolism. We conclude that menin maintains the T-cell functions by limiting mTORC 1 activity and subsequent cellular metabolism.

## Results

**Dysfunction in *menin*-deficient activated CD8 T cells**. We previously reported that *menin* knockout (KO; *menin*^flox/flox mice with *CD4-Cre* transgenic) naive CD4 T cells more rapidly senesced after receiving TCR stimulation than did the wild-type (WT) control cells[10]. Similar to *menin* KO naive CD4 T cells, the growing rate of *menin* KO naive CD8 T cells was reduced after day 7, even in the presence of exogenous IL-2 (Supplementary Fig. 1a). To assess the effects of *menin* deficiency on the cell cycle, we measured the percentage of replicating cells after incubation with 5-ethynyl-2′-deoxyuridine (EdU). A reduced number of EdU-positive cells was also detected in the *menin* KO CD8 T-cell cultures on days 7 (Supplementary Fig. 1b). The proportion of cell death was not increased in the *menin* KO CD8 T-cell cultures (Annexin V positive: approximately 14.1%) compared with that in WT cultures (Annexin V positive: 13.4%) (Supplementary Fig. 1c). The numbers of CD27^low/CD62L^low and CD27^high/CD62L^low cells were markedly increased in the *menin* KO CD8 T-cell cultures compared with those observed in the WT cell cultures (Fig. 1a and Supplementary Fig. 1d). Furthermore, the increased expression of inhibitory receptors, such as PD-1 was detected in the *menin* KO CD8 T-cell cultures (Fig. 1b and Supplementary Fig. 1e). In sharp contrast, the expression of CD226, an activating receptor, was reduced in *menin* KO CD8 T cells (Supplementary Fig. 1e). Moreover, the SASP-like feature was also induced in *menin*-deficient activated CD8 T cells. The number of OPN-producing cells was markedly increased in the *menin* KO CD8 T-cell cultures, whereas the generation of IFN-γ-producing cells remained unaffected (Fig. 1c and Supplementary

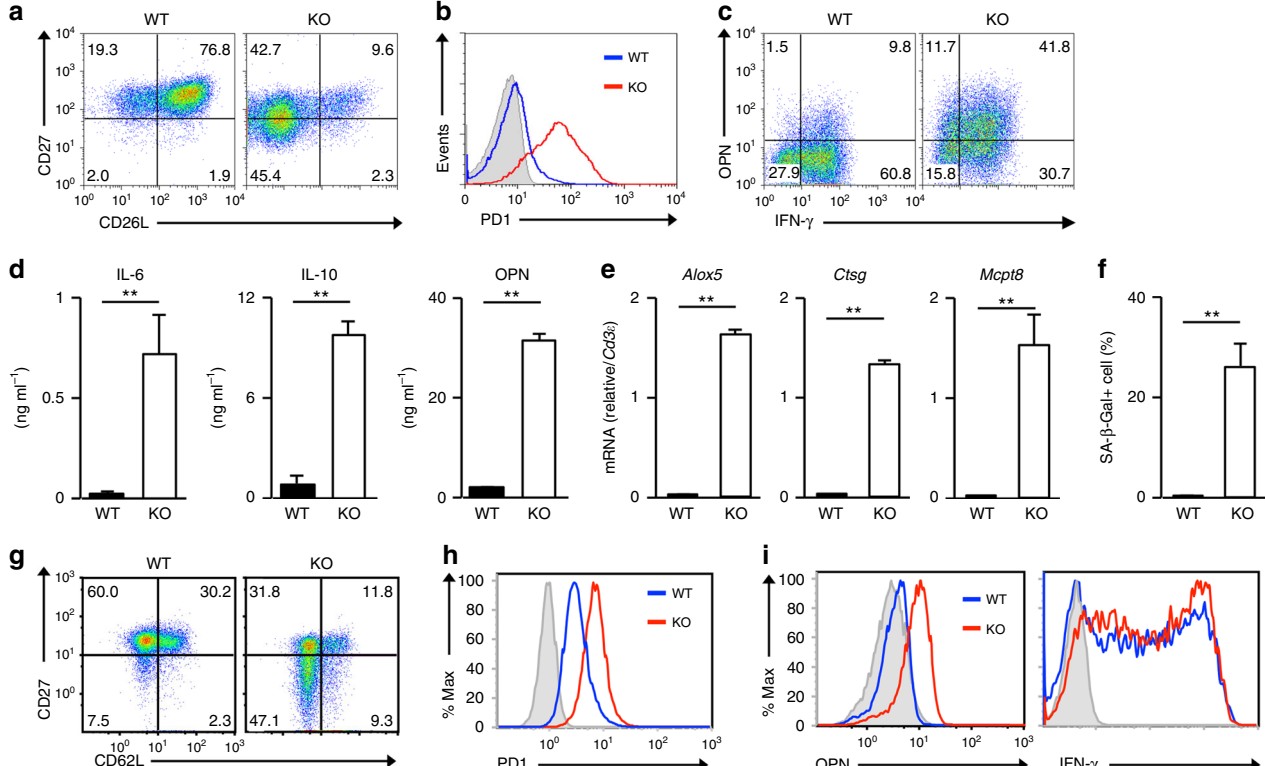

**Fig. 1** Menin deficiency induces dysfunction of CD8 T cells. **a** A representative staining profile of CD62L/CD27 on the cell surface of the WT and *menin* KO effector CD8 T cells. Naive CD8 T cells were stimulated with anti-TCR-β mAb plus anti-CD28 mAb with IL-2 for 2 days, and then the cells were further expanded with IL-2 for an additional 5 days. An analysis was performed on day 7 after the initial stimulation. **b** A representative staining profile of PD-1 on the cell surface of the WT and *menin* KO CD8 T cells on day 7. **c** Representative results of the intracellular FACS analysis of IFN-γ/OPN in the WT and *menin* KO effector CD8 T cells on day 7. The percentages of cells are indicated in each quadrant. **d** The results of ELISA for IL-6, IL-10, and OPN in the supernatants of the cells in **c** restimulated with immobilized anti-TCR-β for 16 h are shown with the standard deviation ($n = 3$: biological replicates). **e** The results of the quantitative RT-PCR analysis of mRNAs encoding pro-inflammatory enzymes in the WT and *menin* KO effector CD8 T cells on day 7. The results are presented relative to the mRNA expression of *Cd3ε* with the standard deviations ($n = 3$: technical replicates). **f** The percentages of SA β-galactosidase (SA β-Gal)-positive cells on day 12 are shown with the standard deviation ($n = 3$: biological replicates). **g** A representative staining profile of CD62L/CD27 on the cell surface of the WT and *menin* KO OT1 Tg splenic CD8 T cells on day 7 after *Lm*-OVA infection. **h** A representative staining profile of PD-1 on the cell surface of the cells in **g**. **i** Representative results of the intracellular FACS analysis of IFN-γ/OPN in the cells in **g** stimulated with an OVA-peptide (SIINFEKL) for 6 h. $**p < 0.01$ (Student's *t*-test)

Fig. 1f). A striking increase in IL-6, IL-10, and OPN production in the *menin* KO CD8 cells was detected using enzyme-linked immunosorbent assays (Fig. 1d). The augmented expression of the pro-inflammatory chemokines (*Ccl2* and *Ccl5*), pro-inflammatory enzymes (*Alox5*, *Ctsg*, *Mcpt8*, and *Mmp13*) and pro-angiogenic factors (*Pdgfα* and *Vegfc*) was detected in *menin* KO effector CD8 T cells (Fig. 1e and Supplementary Fig. 1g). The strong expression of the SA β-Gal activity was detected in the *menin* KO effector CD8 T cells on day 12 (Fig. 1f and Supplementary Fig. 1h). The dysfunction was detected at least 3 days after the initial TCR stimulation in *menin* KO CD8 T cells, whereas this phenotype was not observed in WT CD8 T cells even by on day 12 (Supplementary Fig. 2). We found that these features were not detected in *menin* KO naive CD8 T cells (Supplementary Fig. 2). However, dysfunction was detected in *menin* KO CD8 T cells under stimulation with low-dose anti-TCR-β/anti-CD28 mAb. (Supplementary Fig. 3). Furthermore, a similar phenotype was detected in vivo in *menin* KO CD8 T cells on day 7 after infection with OVA-peptide expressing *Listeria monocytogenes* (*Lm*-OVA). A decreased CD62L and CD27 expression (Fig. 1g and Supplementary Fig. 4a) and increased PD-1 (Fig. 1h and Supplementary Fig. 4b) level were detected in CD8 T cells from *Lm*-OVA-infected *menin*-deficient mice. The

expression of OPN was increased in CD8 T cells from *Lm*-OVA-infected *menin*-deficient mice compared with WT mice, whereas the IFN-γ expression was comparable between the two mouse strains (Fig. 1i and Supplementary Fig. 4c). These results indicate that *menin* KO CD8 T cells rapidly malfunction after receiving TCR stimulation.

**Menin KO CD8 T cells rapidly acquire effector functions.** It was previously reported that menin localizes in the membrane compartment and inhibits Akt activation[54]. The level of menin protein in the cytosolic fraction of aged activated CD8 T cells was lower than that in young cells, whereas the level in the nuclei was comparable (Supplementary Fig. 5). We assessed the effect of menin deficiency on the Akt signaling in activated CD8 T cells. The amount of phosphorylated (Ser473 and Thr308) was increased in *menin* KO activated CD8 T cells compared with those in WT CD8 T cells (Fig. 2a). The phosphorylation of Akt substrates was also increased in *menin* KO activated CD8 T cells (Supplementary Fig. 6a). Furthermore, the phosphorylation of mechanistic target of rapamycin (mTOR) (Ser2448 and Ser2481) (Fig. 2b) and ribosomal protein S6 (Ser235/236 and Ser240/244) (Fig. 2c) was enhanced in *menin* KO activated CD8 T cells. The

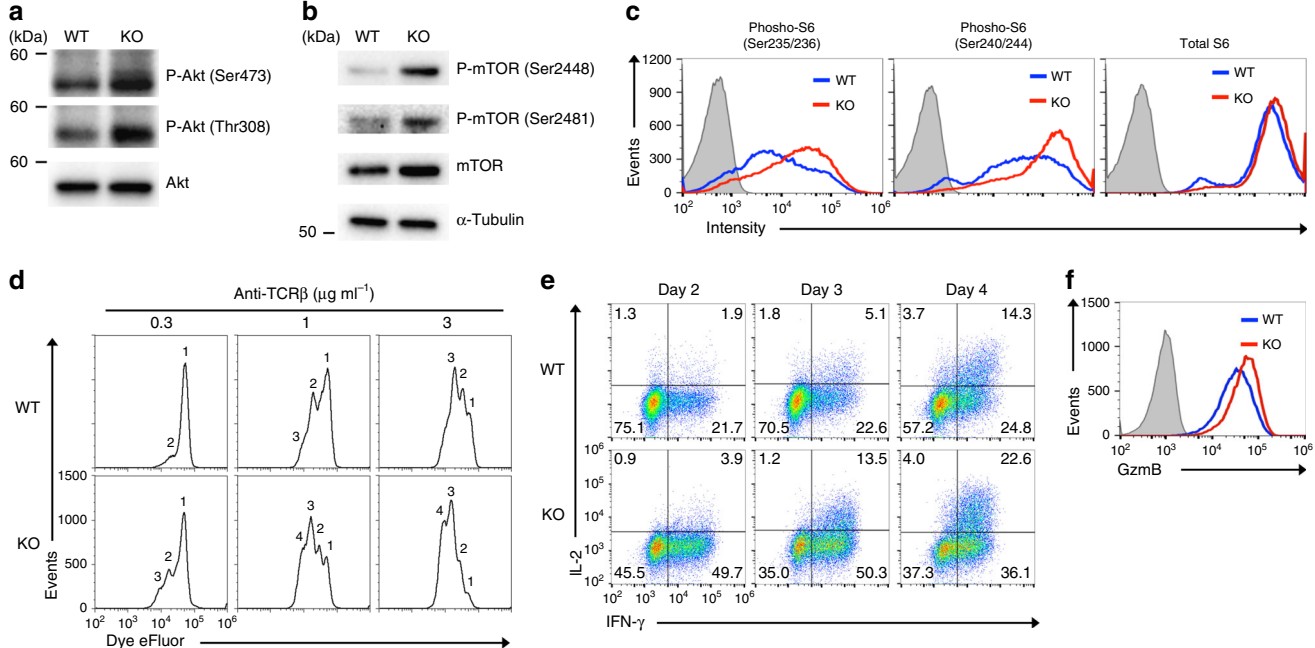

**Fig. 2** *Menin* KO CD8 T cells rapidly proliferate and acquire effector functions. **a** The results of the immunoblot analysis of the phospho-Akt (Ser473 or Thr308) and total Akt protein in the WT or *menin* KO activated CD8 T cells. WT or *menin* KO naive CD8 T cells were stimulated with anti-TCR-β mAb plus anti-CD28 mAb for 36 h and subjected to immunoblotting. **b** The results of the immunoblot analysis of the phospho-mTOR (Ser2448/2481), total mTOR protein and α-tubulin (control) of the cells in **a**. **c** The result of the flow cytometry analysis of phospho-ribosomal S6 (Ser235/236 and Ser240/244) and total ribosomal S6 protein in the cells in **a**. **d** The cell division between eFluor670-labeled WT or *menin* KO naive CD8 T cell was compared upon the indicated concentration of anti-TCR-β mAb plus anti-CD28 mAb in the presence of IL-2 for 48 h. **e** Naive CD8 T cells from the spleen of the WT and *menin* KO mice were stimulated with anti-TCR-β mAb plus anti-CD28 mAb in the presence of IL-2 for 2 days. The cells were then further expanded with IL-2 for the indicated days. Representative results of the intracellular FACS analysis of IL-2/IFN-γ in the WT and *menin* KO CD8 T cells on the indicated days. The percentages of cells are indicated in each quadrant. **f** The results of the intracellular FACS analysis of GzmB in the WT and *menin* KO CD8 T cells on day 3

enhanced phosphorylation of S6 protein in *menin* KO CD8 T cells was also detected in vivo 48 h after *Lm*-OVA-infection (Supplementary Fig. 6b). As indicated, the Akt-mTOR complex signaling plays an important role in regulating the proliferative response and effector functions of T cells. *Menin* KO naive CD8 T cells adequately responded to suboptimal-dose TCR stimulation and divided more quickly than control WT cells after TCR stimulation (Fig. 2d). Furthermore, *menin* KO activated CD8 T cells rapidly acquired the ability to produce IFN-γ and IL-2 production (Fig. 2e). The expression of granzyme B in *menin*-deficient activated CD8 T cells was higher than in WT CD8 T cells (Fig. 2f). These results indicate the augmented activity of the Akt-mTOR complex 1 (mTORC1) signaling in *menin* KO activated CD8 T cells.

**Rapamycin prevents dysfunction in *menin* KO effector CD8 T cells**. We assessed whether or not the increased mTORC1 activity in *menin* KO activated CD8 T cells is involved in dysfunction. Rapamycin, initially identified as an antifungal metabolite produced by *Streptomyces hygroscopics*, specifically suppresses the activity of mTORC1. Rapamycin was added during TCR stimulation, and then cells were expanded with IL-2 without rapamycin for the indicated number of days. The decreased expression of CD62L and CD27 in *menin* KO CD8 T cells was restored by rapamycin (Fig. 3a and Supplementary Fig. 7a). The expression of CD62L in WT activated CD8 T cells was also enhanced by rapamycin treatment. In addition, rapamycin restored the dysregulated cell surface expression of PD-1 and CD226 in *menin* KO CD8 T cells (Fig. 3b). The increased production of IL-6, IL-10, and OPN (Fig. 3c) and the generation of

OPN-producing cells (Supplementary Fig. 7b) in *menin* KO effector CD8 T cells was partially restored by rapamycin treatment. The increased mRNA expression of *Alox5*, *Ctsg*, and *Mcpt8* (Fig. 3d) and increased SA β-Gal activity (Fig. 3e and Supplementary Fig. 7c) in *menin* KO CD8 T cells were normalized by rapamycin.

The decreased formation of immunological memory is a hallmark of immunosenescence. We previously reported the reduction in the number of memory precursor effector (CD127[high]KLRG1[low]) *menin*-deficient CD8 T cells during primary *Lm*-OVA infection and a reduction in the generation of memory CD8 T cells against OVA-peptide in *menin* KO mice[55]. Therefore, we assessed the secondary immune response of *menin*-deficient CD8 T cells against *Lm*-OVA infection by adoptive transfer experiments in vivo (Supplementary Fig. 8a). In vitro-activated effector CD8 T cells with OT1 transgenic (Tg) (Thy1.1[+]), *menin* KO OT1 Tg (Thy1.2[+]) or *menin* KO OT1 Tg (Thy1.2[+]) treated with rapamycin background were mixed at a 1:1 ratio and adoptively transferred into naive C57/BL6 mice (Thy1.1[+]x Thy1.2[+]). Twenty days after adoptive transfer, the mice were infected with *Lm*-OVA, and the number of OT1 Tg CD8 T cells was measured. The impaired secondary immune response against *Lm*-OVA in *menin* KO CD8 T cells was restored by rapamycin treatment during primary TCR stimulation (Fig. 3f and Supplementary Fig. 8b, c). Improvement of the secondary immune response of *menin* KO CD8 T cells by rapamycin was confirmed by adoptive transfer experiments of memory CD8 T cells (Supplementary Fig. 9a). The impaired secondary immune response against *Lm*-OVA in *menin* KO CD8 T cells was also restored by rapamycin treatment during primary TCR stimulation under these conditions (Fig. 3g and Supplementary Fig. 9b,

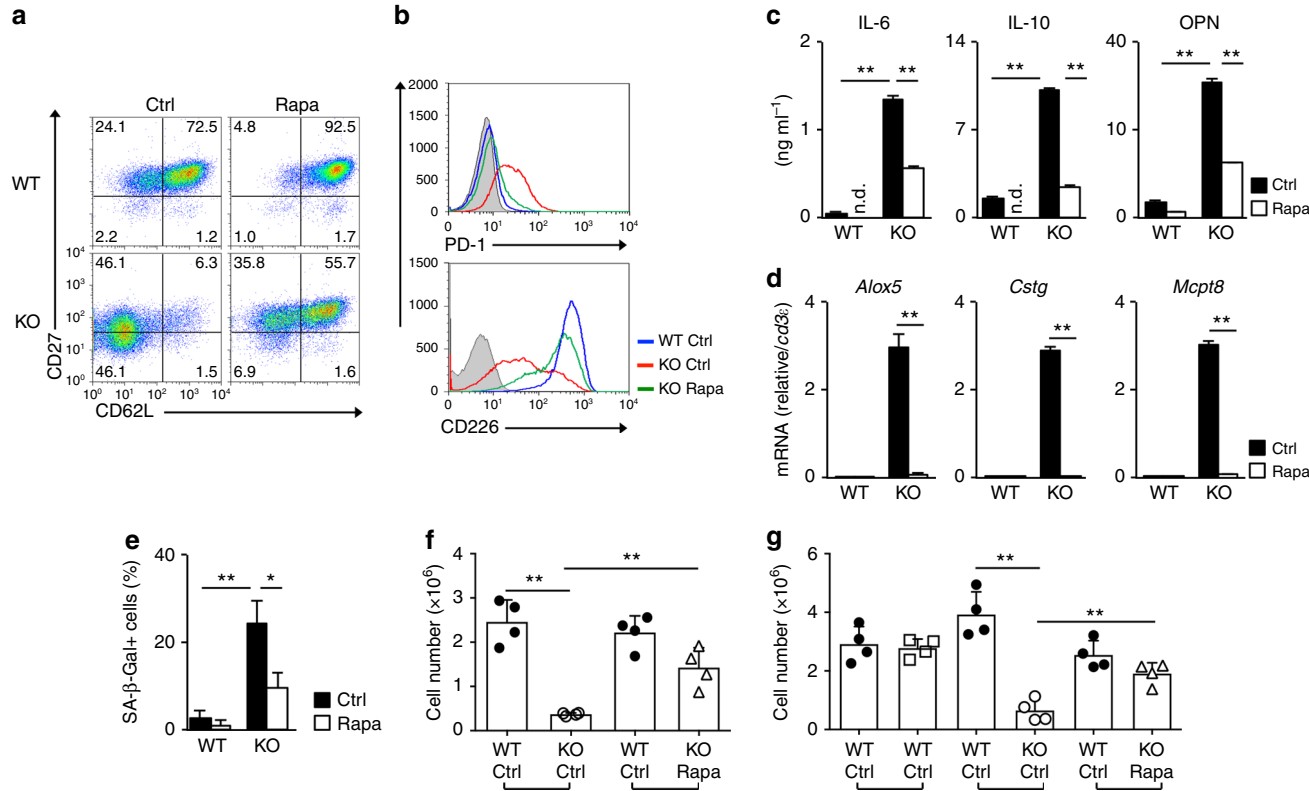

**Fig. 3** Rapamycin inhibits dysfunction of *menin* KO CD8 T cells. **a** A representative staining profile of CD62L/CD27 on the cell surface of the WT and *menin* KO effector CD8 T cells on day 7. The percentages of cells are indicated in each quadrant. Naive CD8 T cells were stimulated with anti-TCR-β mAb plus anti-CD28 mAb with IL-2 in the presence or absence of rapamycin for 2 days, and then the cells were further expanded with IL-2 in the absence of rapamycin for an additional 5 days. **b** Representative staining profiles of CD226 and PD-1 on the surface of the cells in **a**. **c** The ELISA for IL-6, IL-10, and OPN in the supernatants of the cells in **a** restimulated with immobilized anti-TCR-β for 16 h are shown with the standard deviation ($n = 3$: biological replicates). **d** The results of the quantitative RT-PCR analysis of the pro-inflammatory enzymes in the cells in **a**. The results are presented relative to the expression of *Cd3ε* mRNA with the standard deviations ($n = 3$: technical replicates). **e** The percentages of SA β-galactosidase (SA β-Gal)-positive cells on day 12 are shown with the standard deviation ($n = 3$: biological replicates). **f** A 1:1 mixture of WT OT-1 Tg effector CD8 T (Thy1.1+)/*menin* KO OT-1 Tg effector CD8 T cells (Thy1.2+) or WT (Thy1.1+)/rapamycin-treated *menin* KO (Thy1.2+) was adoptively transferred into WT congenic (Thy1.1+ Thy1.2+) mice. Twenty days after the transfer, the mice were infected with *Lm*-OVA to activate the donor cells. The donor cells were collected from the spleen on day 5 after *Lm*-OVA infection and analyzed by FACS. The absolute number of donor cells in the spleen was indicated (mean ± SD, $n = 4$ per group: biological replicates). **g** WT (Thy1.1+ or Thy1.2+), *menin* KO or rapamycin-treated *menin* KO OT-1 Tg memory CD8 T cells (Thy1.2+) were mixed and transferred into WT congenic mice (Thy1.1+ Thy1.2+) as in **f**. The mice were infected with *Lm*-OVA the next day and analyzed as in **f**. The absolute number of donor cells in the spleen is shown (mean ± SD, $n = 4$ per group: biological replicates). *$p < 0.05$, **$p < 0.01$ (Student's *t*-test)

c). These results suggest that the increased activity of mTORC1 during initial TCR stimulation is involved in the induction of dysfunction in *menin* KO CD8 T cells.

**Enhanced central carbon metabolism in *menin* KO CD8 T cells.** Since mTOR signaling is a metabolic cue for proper maintenance and activation of T cells[22,23], we next wanted to determine whether or not menin regulates the metabolic process in activated CD8 T cells. The incorporation of the glucose analog 2-NBDG (2-deoxy-2-[(7⁻nitro-2,1,3-benzoxadiazol-4-yl)amino]-D-glucose) was higher in *menin* KO activated CD8 T cells than in WT CD8 T cells (Fig. 4a). The intracellular level of glutamate was dramatically increased by anti-TCR-β plus anti-CD28 mAb stimulation, and the level was significantly higher in *menin* KO activated CD8 T cells than in WT CD8 T cells (Fig. 4b). Both the incorporation of 2-NBDG (Supplementary Fig. 10a) and upregulation of the intracellular glutamate (Supplementary Fig. 10b) in activated CD8 T cells were partially inhibited by rapamycin indicating that mTORC1 signaling is involved in the TCR-mediated activation of central carbon metabolism. To detect the

metabolic status in WT and *menin*-deficient CD8 T cells comprehensively, naive CD8 T cells were stimulated with an anti-TCR-β mAb plus anti-CD28 mAb for 36 h and subjected to metabolic profiling of 116 metabolites. The intracellular concentration of glycolytic metabolites such as glucose 6-phosphate and fructose 1,6-diphosphate was increased in *menin* KO CD8 T cells (Fig. 4c). The level of lactate, an end-product of anaerobic glycolysis, also increased in *menin* KO CD8 T cells, while the concentration of pyruvate moderately decreased (Fig. 4c). The intracellular levels of glutamine and glutamate were significantly reduced in *menin* KO CD8 T cells compared to WT CD8 T cells 36 h after the initial stimulation (Fig. 4d), while the concentrations of metabolic intermediates of the TCA cycle such as succinate, fumarate and malate, increased (Fig. 4e). The levels of citrate, *cis*-aconitate, and isocitrate were marginally decreased (Fig. 4e), implying accelerated OXOPHOS via glutaminolysis in *menin* KO CD8 T cells. The increased anaerobic glycolysis (extracellular acidification rates [ECAR]) (Fig. 4f) and oxygen consumption rates (OCR) (Fig. 4g) in *menin* KO activated CD8 T cells was confirmed using a Seahorse Extracellular Flux Analyzer (Agilent Technologies, Santa Clara, CA, USA) under both

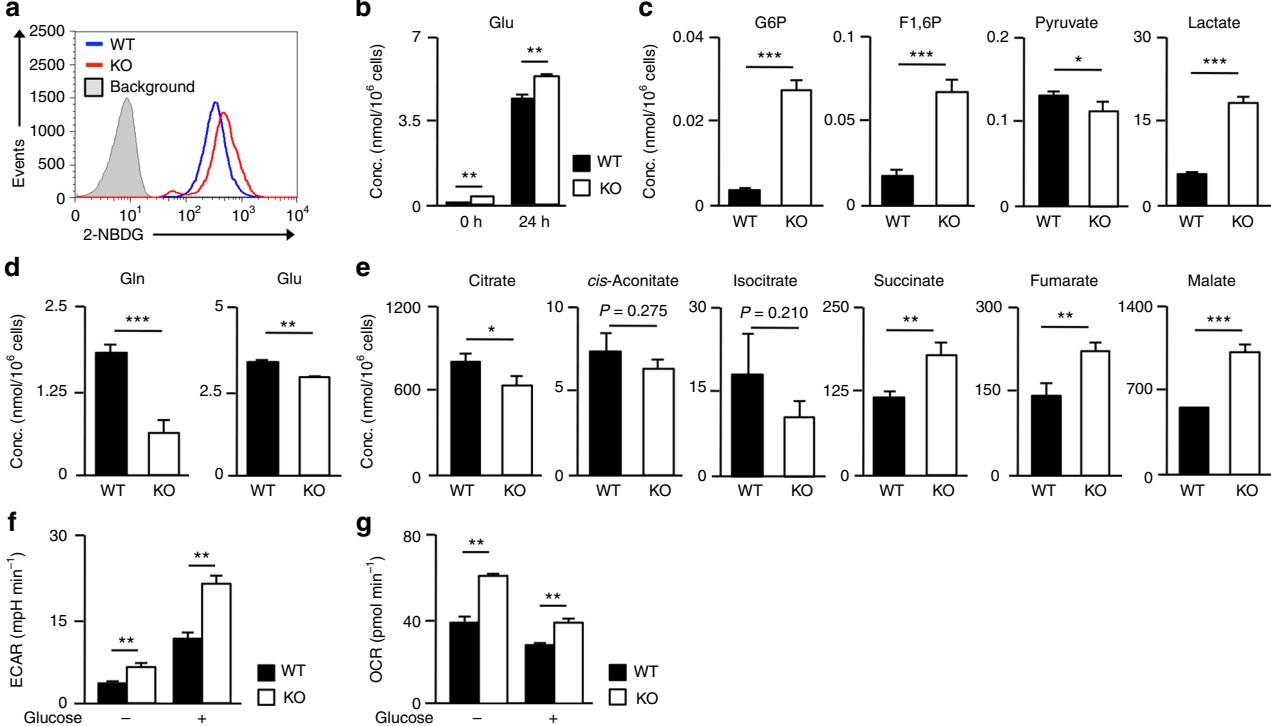

**Fig. 4** Metabolic profiling of *menin* KO activated CD8 T cells. **a** WT or *menin* KO naive CD8 T cells were stimulated with anti-TCR-β and anti-CD28 mAbs in the presence of IL-2 for 24 h. 2-NBDG was then added to the cultures for 30 min, and its incorporation was determined by FACS. A representative FACS profile is shown. **b** The intracellular level of glutamine in naive and activated CD8 T cells with WT or *menin* KO background are indicated ($n = 3$: biological replicates). The results are presented with the standard deviation. **c** The intracellular amount of glucose 6-phosphate (G6P), fructose 1,6-diphosphate (F1, 6P), pyruvate, and lactate in the WT and *menin* KO activated CD8 T cells are presented with the standard deviation ($n = 3$: biological replicates). Naive CD8 T cells were stimulated with anti-TCR-β and anti-CD28 mAbs in the presence of IL-2 for 36 h. **d** The intracellular amount of glutamine and glutamate of the cells in **c**. The results are presented with the standard deviation ($n = 3$: biological replicates). **e** The intracellular amounts of TCA cycle intermediates of the cells in **c**. The results are presented with the standard deviation ($n = 3$: biological replicates). **f, g** Naive CD8 T cells were stimulated with anti-TCR-β and anti-CD28 mAbs in the presence of IL-2 for 36 h, and then glycolysis (**f**) and the OCR (**g**) were determined before or 20 min after glucose (10 mM) injection. **b, f, g** **$p < 0.01$ (Student's *t*-test), **c, d, e** *$p < 0.05$, **$p < 0.01$, ***$p < 0.001$ (Welch's *t*-test)

glucose-deprived and glucose-sufficient conditions. These data indicate that both anaerobic glycolysis and glutaminolysis are facilitated in CD8 T cells by menin deficiency.

**Glutamine-α-KG axis regulates CD8 T-cell dysfunction.** We examined whether or not an enhanced glutamine metabolism is involved in the induction of dysfunction in *menin* KO activated CD8 T cells. To assess the role of glutamine metabolism, we used 6-diazo-5-oxo-L-norleucine (L-Don), a glutamine analog that antagonizes glutamine, and aminooxyacetic acid (AOA), an inhibitor of transaminases. The intracellular concentration of glutamate in activated CD8 T cells decreased by L-Don or AOA treatment (Supplementary Fig. 11a). The decreased expression of CD62L and CD27 and increased expression of PD-1 in *menin* KO CD8 T cells was partially restored by L-Don or AOA treatment (Supplementary Fig. 11b, c). Furthermore, both inhibitors also inhibited the increased SA β-Gal activity in *menin* KO CD8 T cells (Supplementary Fig. 11d).

To directly confirm the role of glutamine and subsequent glutaminolysis on the induction of dysfunction, we stimulated T cells with anti-TCR-β mAb plus anti-CD28 mAb in the presence of IL-2 and cultured the cells for the first three days under glutamine-deprived conditions (culture medium with low L-glutamine). The cells were then cultured in normal medium. The L-glutamine level in this culture medium is reduced (control: 3 mM, glutamine-deprived: 0.05 mM) but not completely depleted, as a substantial amount of L-glutamine is present in fetal calf serum. To confirm the role of α-KG, α-KG analog dimethyl (DM)- α-KG was

added into the glutamine-deprived cultures. The intracellular concentrations of glutamine and glutamate were significantly reduced in CD8 T cells cultured under glutamine-deprived conditions for 36 h (Supplementary Fig. 12a). The decreased expression of CD62L and CD27 in *menin* KO CD8 T cells was normalized by glutamine deprivation, and its effect was antagonized by DM-α-KG administration (Fig. 5a and Supplementary Fig. 12b). The expression of CD62L was also enhanced in WT activated CD8 T cells cultured under glutamine-deprived conditions. The cell surface expression of CD226 and PD-1 in *menin* KO effector CD8 T cells was normalized by glutamine deprivation and its effect was inhibited by DM-α-KG (Fig. 5b). The increased production of IL-6, IL-10, and OPN and the enhanced expression of the pro-inflammatory enzymes in *menin* KO CD8 T cells were suppressed by glutamine deprivation (Fig. 5c, d). The effects of glutamine deprivation were antagonized again by the administration of DM-α-KG (Fig. 5c, d). The increased SA β-Gal activity in *menin* KO CD8 T cells was also regulated by glutamine and DM-α-KG (Fig. 5e and Supplementary Fig. 12c). The impairment of the secondary response against *Lm*-OVA in *menin* KO CD8 T cells was partially restored by glutamine deprivation during primary TCR stimulation (Fig. 5f and Supplementary Fig. 12d). Furthermore, the secondary immune response against *Lm*-OVA in WT CD8 T cells cultured under glutamine-deprived conditions was significantly increased, and the effect of glutamine deprivation was antagonized by the addition of DM-α-KG (Supplementary Fig. 12e). These results suggest that glutamine-α-KG axis participates in the dysfunction and the fate decision of activated CD8 T cells.

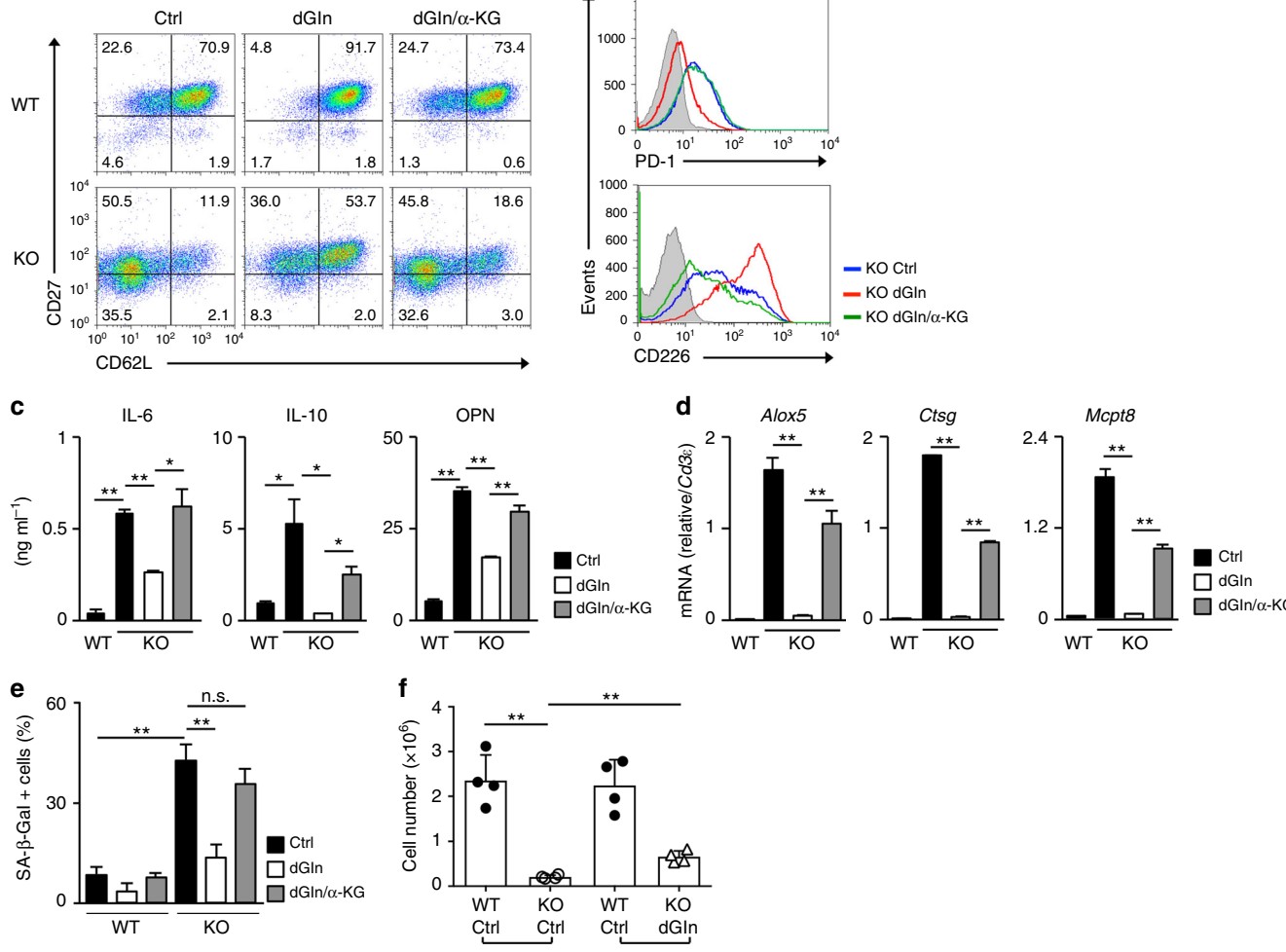

**Fig. 5** The glutamine-α-KG axis is involved in the dysfunction in *menin* KO CD8 T cells. **a** A representative staining profile of CD62L/CD27 on the cell surface of the WT and *menin* KO effector CD8 T cells on day 7. The percentages of cells are indicated in each quadrant. Naive CD8 T cells were activated and cultured for 3 days under normal (Ctrl), glutamine-deprived (dGln), or glutamine-deprived supplemented with DM-α-KG (dGln/α-KG) conditions for 3 days, and then the cells were further expanded with IL-2 under normal conditions for an additional 4 days. An analysis was performed on day 7 after the initial anti-TCR-β/CD28 stimulation. **b** Representative staining profiles of CD226 and PD-1 on the cell surface of the cells in **a**. **c** The ELISA results for IL-6, IL-10, and OPN in the supernatants of the cells in **a** restimulated with immobilized anti-TCR-β for 16 h are shown with standard deviation ($n = 3$: biological replicates). **d** The results of the quantitative RT-PCR analysis of the pro-inflammatory enzymes in the cells in **a**. The results are presented relative to the expression of *Cd3ε* mRNA with the standard deviations ($n = 3$: technical replicates). **e** The percentages of SA β-galactosidase (SA β-Gal)-positive cells on day 12 after the initial anti-TCR-β/CD28 mAb stimulation are shown with the standard deviation ($n = 3$: biological replicates). **f** A 1:1 mixture of WT OT-1 Tg effector CD8 T (Thy1.1+)/*menin* KO OT-1 Tg effector CD8 T cells under normal conditions (Thy1.2+) or WT (Thy1.1+)/*menin* KO under glutamine-deprived conditions (Thy1.2+) was adoptively transferred into WT congenic (Thy1.1+ Thy1.2+) mice. Twenty days after the transfer, the mice were infected with *Lm*-OVA to activate the donor cells. The donor cells were collected from the spleen on day 5 after *Lm*-OVA infection and analyzed by FACS. The absolute number of donor cells in the spleen is shown (mean ± SD, $n = 4$ per group: biological replicates). *$p < 0.05$, **$p < 0.01$ (Student's *t*-test)

**Histone H3K27 demethylation is involved in CD8 T-cell dysfunction.** It has been well established that α-KG acts as a cofactor of histone and DNA demethylases[56,57]. We found that the levels of histone H3K27 di-methylation (me2) and H3K27 tri-methylation (me3) in activated CD8 T cells were more sensitive to the glutamine and α-KG concentration than were the levels of H3K4me3. The levels of histone H3K27me2 and H3K27me3 in activated CD8 T cells increased under glutamine-deprived conditions and reduced in the presence of α-KG, whereas the H3K4me3 level was unaffected (Fig. 6a). The levels of histone H3K27me2 and me3 moderately decreased in *menin* KO activated CD8 T cells compared to WT activated CD8 T cells (Fig. 6b).

Two kinds of histone H3K27me2/me3 demethylases have been identified: Kdm6a (Utx) and Kdm6b (Jmjd3)[58,59]. The H3K27 demethylase UTX-1, an orthologue of mammalian Utx, regulates

the lifespan of *Caenorhabditis elegance*[60,61]. Utx-1 causes a genome-wide decrease in the histone H3K27 tri-methylation in response to insulin/IGF-1 signaling and promotes aging. We, therefore, generated *utx*-floxed mice (Supplementary Fig. 13a) and established T-cell-specific *menin/utx*-double deficient (*menin*flox/flox x *utx*flox/flox x CD4-Cre Tg) mice to assess the role of H3K27me2/me3 demethylation in CD8 T-cell senescence. The expression of *utx* mRNA was not detected in *utx*-deficient activated CD8 T cells (Supplementary Fig. 13b). The decreased expression of CD62L and CD27 in *menin* KO CD8 T cells was partially restored by utx deficiency (Fig. 6c and Supplementary Fig. 13c). The expression of CD62L was also enhanced in *utx* KO activated CD8 T cells (Fig. 6c and Supplementary Fig. 13c). The deletion of the *utx* reduced the cell surface expression of PD-1 in *menin* KO effector CD8 T cells (Fig. 6d). The increased

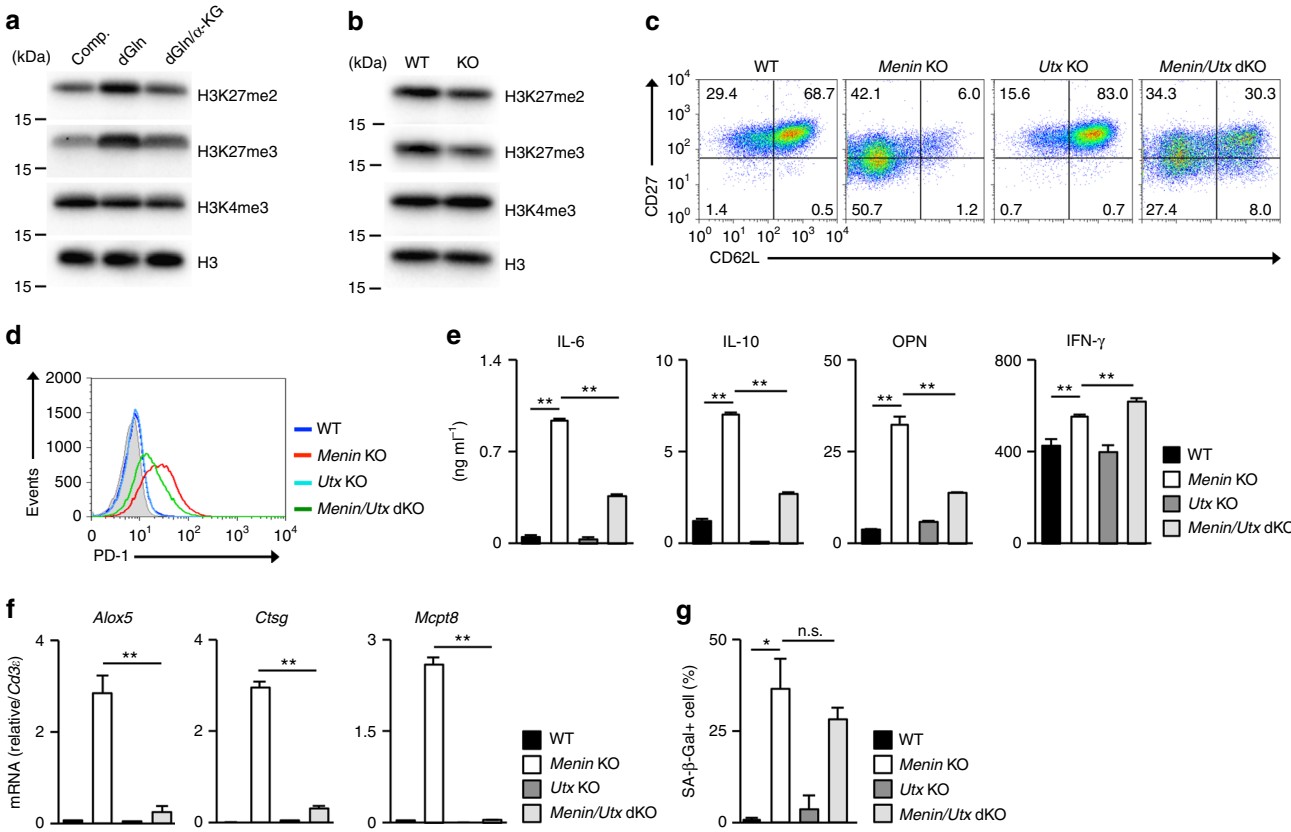

**Fig. 6** Demethylation of histone H3K27 is involved in CD8 T-cell dysfunction. **a** The result of the immunoblot analysis of the di- or tri-methylated histone H3K27, tri-methylated histone H3K4 and total histone H3 in the WT CD8 T cells cultured under the indicated conditions for 48 h. The protein amount of histone H3 was used as a loading control. **b** The results of the immunoblot analysis of the di- or tri-methylated histone H3K27, tri-methylated histone H3K4 and total histone H3 in the WT or *menin* KO CD8 T cells cultured under normal conditions for 3 days. **c** A representative staining profile of CD62L/CD27 on the cell surface of the WT, *utx* KO *menin* KO or *menin/utx*-double KO effector CD8 T cells on day 7. The percentages of cells are indicated in each quadrant. **d** Representative staining profile of PD-1 on the cell surface of the cells in **c**. **e** The ELISA results for IL-6, IL-10, OPN, and IFN-γ in the supernatants of the cells in **c** restimulated with immobilized anti-TCR-β for 16 h are shown with the standard deviation (n = 3: biological replicates). **f** The results of the quantitative RT-PCR analysis of the pro-inflammatory enzymes of the cells in **c**. The results are presented relative to the expression of *Cd3ε* mRNA with the standard deviations (n = 3: technical replicates). **g** The percentages of SA β-galactosidase (SA β-Gal)-positive cells on day 12 after the initial anti-TCR-β/CD28 stimulation are shown with the standard deviation (n = 3: biological replicates). *p < 0.05, **p < 0.01 (Student's t-test)

production of IL-6, IL-10, and OPN in *menin* KO effector CD8 T cells was partially suppressed by the utx deficiency, whereas the IFN-γ production was moderately increased (Fig. 6e). The increased mRNA expression of SASP factors, such as *Alox5*, *Ctsg*, and *Mcpt8* in *menin* KO CD8 T cells was normalized by the *utx* deficiency (Fig. 6f). In sharp contrast, the increased SA β-Gal activity in *menin* KO CD8 T cells was not inhibited by the *Utx* deficiency (Fig. 6g and Supplementary Fig. 13d). These results suggest that the α-KG-dependent demethylation of histone H3K27 is involved in the dysfunction in *menin* KO CD8 T cells.

**α-KG activates the mTORC1 and central carbon metabolism.** Finally, we examined the impact of α-KG administration on the mTORC1 activity and metabolism in effector CD8 T cells, since the α-KG-mediated activation of mTORC1 has been reported in other cell lines[62]. The phosphorylation of mTOR (Ser2448) (Fig. 7a) and ribosomal S6 protein (Ser240/244) (Fig. 7b) in activated CD8 T cells was reduced under glutamine-deprived conditions and restored by DM-α-KG. The phosphorylated and total AMPKα level decreased in activated CD8 T cells under glutamine-deprived conditions, suggesting that the deprivation of glutamine did not induce AMPK activation in activated CD8 T cells (Supplementary Fig. 14). Furthermore, the administration of α-KG marginally enhanced the

phosphorylated and total AMPKα level. These results indicated that glutamine-α-KG axis controls mTOR phosphorylation through AMPK-independent pathway. We found that DM-α-KG stimulated both ECAR (Fig. 7c) and OCR (Fig. 7d) in effector CD8 T cells cultured under glutamine-deprived conditions. The incorporation of 2-NBDG was reduced by glutamine deprivation and also restored by DM-α-KG administration (Supplementary Fig. 15a). The intracellular concentrations of glycolytic intermediates such as fructose 1,6-diphosphate (F1, 6P) and 3-phosphoglycerate (3-PG) were reduced by glutamine deprivation, and decreased levels of F1, 6P, and 3-PG under glutamine-deprived conditions were recovered by DM-α-KG addition (Supplementary Fig. 15b). The intracellular concentration of succinate, an intermediate of the TCA cycle, was also reduced by glutamine deprivation and restored by DM-α-KG (Supplementary Fig. 15b). Furthermore, the intracellular concentration of NADH and the NADH/NAD+ ratio, an indicator of the central carbon metabolism activity, were markedly reduced in the activated CD8 T cells under glutamine-deprived conditions and normalized by DM-α-KG (Supplementary Fig. 15c), suggesting the α-KG-dependent regulation of the central carbon metabolism in activated CD8 T cells. The expression of glucose transporter (*Slc2a1*), glycolytic enzymes (*Hk2*, *Gapdh*, and *Ldha*) and gluta-minolytic enzymes (*Got1*, *Psat1*) was also decreased by glutamine deprivation in activated CD8 T cells, and the effect was antagonized

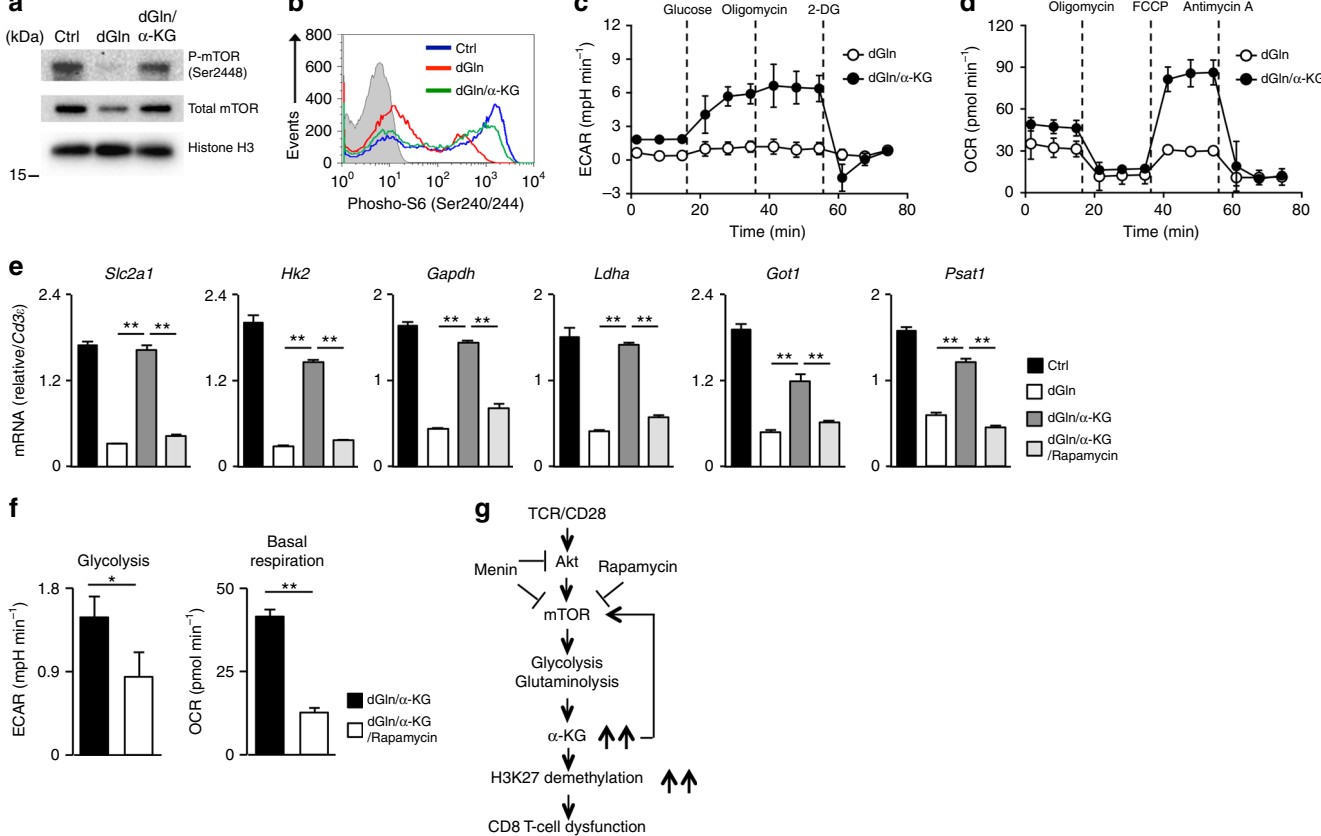

**Fig. 7** α-KG induces the activation of the central carbon metabolism. **a** The results of the immunoblot analysis of phospho-mTOR (Ser2448), mTOR and histone H3 in CD8 T cells cultured under the indicated conditions for 48 h. The protein amount of histone H3 was used as a loading control. **b** DM-α-KG-dependent induction of the phosphorylation (Ser240/244) of ribosomal S6 protein in CD8 T cells stimulated for 24 h. **c**, **d** Naive CD8 T cells were stimulated with anti-TCR-β, anti-CD28 mAbs plus IL-2 without glutamine in the presence or absence of DM-α-KG for 36 h and then ECAR (**c**) and the OCR (**d**) were determined. **e** The results of the quantitative RT-PCR analysis of the metabolic enzymes in the CD8 T cells cultured under as in the indicated conditions for 36 h. The results are presented relative to the expression of *Cd3ε* mRNA with the standard deviation ($n = 3$: technical replicates). **f** Naive CD8 T cells were stimulated with anti-TCR-β, anti-CD28 mAbs plus IL-2, and DM-α-KG without glutamine in the presence or absence of rapamycin (10 nM) for 36 h, and then the ECAR (left) and the OCR (right) were determined. Glycolysis was calculated by subtracting the ECAR before glucose injection from the ECAR 20 min after injection. Basal mitochondria respiration was calculated by subtracting the OCR in the presence of rotenone/antimycin A from the basal OCR at 15 min after the start of measurement. **g** Menin prevents effector CD8 T-cell dysfunction by targeting mTORC1-dependent metabolic activation. **e**, **f** *$p < 0.05$, **$p < 0.01$ (Student's *t*-test)

by DM-α-KG (Fig. 7e). Interestingly, the α-KG-dependent induction of these enzymes was also inhibited by rapamycin (Fig. 7e). In addition, we found that α-KG-induced augmentation of glycolysis (Fig. 7f, left) and basal respiration (Fig. 7f, right) under glutamine-deprived conditions was inhibited by rapamycin.

In conclusion, menin limits mTORC1-dependent metabolic activation, and the loss of menin induces sustained activation of the central carbon metabolism, including glycolysis and glutaminolysis. α-KG production via metabolic activation and subsequent α-KG-dependent histone H3K27 demethylation seems to play a critical role in the induction of CD8 T-cell dysfunction. Furthermore, α-KG may activate mTORC1 signaling, with α-KG and mTORC1 forming a feedback loop to sustain the activation of the central carbon metabolism (Fig. 7g).

## Discussion

In this study, we demonstrated the critical role of menin in regulating the mTORC signaling and cellular metabolism. The menin-dependent restriction of mTORC1 activity and cellular metabolism during the initial TCR-mediated activation phase seem to be important for preventing the induction of dysfunction in activated CD8 T cells. We found that the glutamine-α-KG axis plays a critical

role in inducing CD8 T-cell dysfunction. The glutamine-α-KG axis and mTORC1 seem to form a positive feedback loop to sustain the central carbon metabolism. mTORC1 activity is required for the activation of glycolysis and glutaminolysis, while glutamine and α-KG are required for the sustained activation of mTORC1 in activated CD8 T cells. These results imply that the level of glutamine metabolism during the antigen recognition phase may determine the CD8 T-cell fate through modulation of the mTORC1 activity. Our present data, therefore, suggest the critical role of menin in regulating the cellular metabolism and the subsequent fate decision of activated CD8 T cell.

It was reported that 2-hydroxyglutarate (2-HG), a metabolite of α-KG, activates the mTOR-signaling pathway[63]. 2-HG then leads to the activation of mTOR by decreasing the protein stability of the DEP domain-containing mTOR-interacting protein (DEPTOR), a negative regulator of mTORC1/2. Furthermore, the accumulation of R- and S-2-HG in CD8 T cells after receiving TCR stimulation was reported[64]. The S-2-HG produced in antigen-stimulated CD8 T cells appears to be derived from extracellular glutamine[64] and is produced by lactate dehydrogenase A (Ldha) and/or malate dehydrogenase[65]. We demonstrated that the expression of *Ldha* was reduced by glutamine deprivation and induced in an α-KG-

dependent manner in activated CD8 T cells. We also showed that the ECAR was increased by α-KG in CD8 T cells cultured under glutamine-deprived conditions, indicating the augmented enzymatic activity of Ldha by α-KG. In addition, we found that the concentration of 2-HG was significantly higher in *menin*-deficient activated CD8 T cells than in WT CD8 T cells, and the 2-HG level was reduced by glutamine deprivation and restored by α-KG supplementation in WT activated CD8 T cells (Supplementary Fig. 16). We, therefore, speculated that α-KG activates the mTORC1-signaling pathway in CD8 T cells through enzymatic conversion to 2-HG.

Senescent CD8 T cells are found within the CD27−CD28− population, and these terminally differentiated T cells can be subdivided into two populations: CD45RA+ or CD45RA−. CD27−CD28−CD45RA+ CD8 T cells have multiple features of senescence, including a low proliferative activity and SASP[66]. In addition, one of the most prominent changes in T cells in elderly people is the progressive accumulation of highly differentiated effector memory T ($T_{EM}$) cells (CD45RA+CD28−CD27−CD62L−)[67]. It was reported that human immunodeficiency virus-1 and human cytomegalovirus chronic infection induces CD27−CD28−CD45RA+ senescent CD8 T cells in an age-independent manner[4,5]. These findings suggest that chronic and/or a robust stimulation with antigen and/or cytokine accelerates T-cell senescence in an age-independent manner. We demonstrated that CD62L and CD27 expression was rapidly reduced in *menin* KO CD8 T cells after receiving antigenic stimulation in vivo and in vitro. We also confirmed that some portion of CD62L^low^CD27^low^ *menin* KO CD8 T cells had a down-regulated CD28 expression (Supplementary Fig. 17). Furthermore, the cytosolic menin protein was decreased in aged effector CD8 T cells. Thus, menin may inhibit T-cell senescence, and *menin*-deficient mice may be useful as an antigen-induced as well as aging-induced T-cell senescence model.

α-KG regulates the enzymatic activity of TET2, JmjC family histone demethylases and PDH hydroxylases[45]. We found that α-KG preferentially reduced histone H3K27me2/3 methylation in activated CD8 T cells. The methylation level of histone H3K27me2/3 was decreased in *menin* KO activated CD8 T cells, and the deletion of the *utx* gene partially restored dysfunction in *menin* KO CD8 T cells. These results indicate that utx-dependent histone H3K27 demethylation is involved in the induction of dysfunction in *menin* KO CD8 T cells. It was reported that the H3K27 demethylase UTX-1, an orthologue of mammalian Utx, regulates the lifespan of *Caenorhabditis. elegans*[60,61]. Utx-1 causes a genome-wide decrease in histone H3K27 tri-methylation in response to insulin/IGF-1 signaling and promotes aging. In addition, the insulin/IGF-1 signaling activates mTORC1 through the PI3K-Akt signaling pathway in mammals[68]. Thus, the prolonged activation of the mTORC1 signal in *menin* KO CD8 T cells seems to accelerate the histone H3K27me2/me3 demethylation via the augmentation of glutamine metabolism, thereby inducing dysfunction. We previously reported the prolonged activation of NF-κB, a marker of senescent cells in *menin* KO CD4 T cells[10]. The hypomethylation of histone H3K27 and the augmented activation of NF-κB cooperate to induce dysfunction in *menin* KO T cells.

## Methods

**Mice.** *Menin*^flox/flox^ mice, and Cre TG mice under the control of the *Cd4* promoter were purchased from The Jackson Laboratory. *Utx*^flox/flox^ mice were established by Drs. Kazuki Inoue and Yuuki Imai (Ehime University). Gene-manipulated mice with C57BL/6 background were used in all experiments. C57BL/6 mice purchased from Clea Japan, Inc. (Tokyo, Japan). Female mice were used in the in vivo experiments. Both male and female mice were used in the in vitro experiments. All mice were maintained under specific-pathogen-free conditions and were used at 8–12 weeks of age. All of the animal experiments received

approval from the Ehime University Administrative Panel for Animal Care. All animal care was conducted in accordance with the guidelines of Ehime University. All experiments using *Lm*-OVA were performed in accordance with the protocols approved by the Ehime University Institution Biosafety Committee. The sample size estimate, randomization, and blinding were not used in this study.

**Reagents and antibodies.** Rapamycin was purchased from Wako Chemicals (cat#R0161; Osaka, Japan). Dimethyl alpha-ketoglutarate (DM-αKG) was obtained from Tokyo Chemical Industry (cat#K0013; Tokyo, Japan). Antibodies used for intracellular and cell-surface staining were as follows: anti-Osteopontin-phycoerythrin (PE) mAb (cat#IC808P; R&D Systems, Minneapolis, MN, USA), anti-IFN-γ-allophycocyanin (APC) mAb (cat#554413; BD Bioscience, San Jose, CA, USA), anti-IFN-γ-fluorescein isothiocyanate (FITC) mAb (cat#554411; BD Bioscience), IL-2-APC mAb (cat#JES6-5H4; TONBO Biosciences, San Diego, CA, USA), anti-Phospho-S6 (Ser235/236)-Alexa Fluor 647 mAb (cat#4851; Cell Signaling Technology, Danvers, MA, USA), anti-Phospho-S6 (Ser240/244) Alexa Fluor 647 mAb (cat#5044; Cell Signaling Technology), anti-S6 ribosomal protein Alexa Fluor 647 mAb (cat#5548; Cell Signaling Technology), anti-CD27-PE mAb (cat#558754; BD Bioscience), anti-CD62L-APC mAb (cat#20-0621; TONBO Biosciences), anti-CD226-APC mAb (cat#128809; BioLegend, San Diego, CA, USA), and anti-PD-1-PE mAb (cat#551892; BD Bioscience). The antibodies for immunoblotting were as follows: anti-Akt Ab (cat#9272; Cell Signaling Technology), anti-Phospho-Akt (Ser473) Ab (cat#9271; Cell Signaling Technology), anti-Phospho-Akt (Thr308) Ab (cat#13038; Cell Signaling Technology), anti-Phospho-mTOR (ser2448) pAb (cat#2971; Cell Signaling Technology), anti-mTOR pAb (cat#2983; Cell Signaling Technology), anti-histone H3K4me3 mAb (cat#61379; Active Motif, Carlsbad, CA, USA), anti-histone H3K27me2 pAb (cat#ab39245; Abcam, Cambridge, CA, USA), anti-histone H3K27me3 pAb (cat#39157; Active Motif) and anti-histone H3 mAb (cat#39763; Active Motif), and anti-α-tubulin (cat#2125; Cell Signaling Technology). The antibodies were diluted by 50 times for flow cytometry (FACS) analysis and by 1000 times for immunoblotting, respectively.

**CD8 T-cell stimulation and differentiation in vitro.** Naive CD8 T (CD44^low^CD62L^high^) cells were prepared using a Naive CD8+ T-cell Isolation kit (cat#130-096-543; Miltenyi Biotec, San Diego, CA, USA). Naive CD8 T cells ($1.5 \times 10^6$) were stimulated with immobilized anti-TCR-β mAb (3 μg/ml, H57-597; BioLegend) and anti-CD28 mAb (1 μg/ml, 37.5; BioLegend) for 2 days in the presence of IL-2 (10 ng/ml, Pepro Tech). The cells were then transferred to a new plate and further cultured in the presence of IL-2 (10 ng/ml). The cells were cultured in RPMI 1640 with ʟ-glutamine (cat#189-02025; Wako Chemicals) supplemented with 10% fetal bovine serum, 2 mM ʟ-glutamine (16948-04; Nacalai Tesque, Kyoto, Japan), 1 mM sodium pyruvate (cat#06977-34; Nacalai Tesque), 1% MEM nonessential amino acids (cat#06344-56; Nacalai Tesque), 10 mM HEPES (cat#15630-080; Thermo Fisher Scientific, Waktham, MA, USA), 55 μM 2-Mercaptoethanol (cat#21985-023; Thermo Fisher Scientific), and 1% penicillin-streptomycin (cat#26253-84; Nacalai Tesque). For glutamine-deprived conditions, the cells were cultured in RPMI 1640 without ʟ-glutamine (cat#183-02165; Wako Chemicals) supplemented with 10% fetal bovine serum, 1 mM sodium pyruvate, 1% MEM nonessential amino acids, 10 mM HEPES, 55 μM 2-mercaptoethanol, and 1% penicillin-streptomycin.

**Intracellular staining of cytokines.** The cells were differentiated in vitro and stimulated with an immobilized anti-TCR-β mAb (3 μg/ml, H57-597; BioLegend) for 6 h with monensin (2 μM, cat#M5273; Sigma-Aldrich, St. Louis, MO, USA) for the intracellular staining of cytokines. Intracellular staining was performed as described previously. For the intracellular staining of phosphorylated S6 ribosomal proteins, the CD8 T cells were fixed and permeabilized with BD Phosflow Lyse/Fix Buffer (cat#558049; BD Bioscience) and BD Phosflow Perm III (cat#558050; BD Bioscience) in accordance with the manufacturer's instructions. Flow cytometry was performed using a FACS Caliber instrument (BD Biosciences) and Gallios instrument (Beckman Coulter, CA, USA), and the results were analyzed using the FlowJo software program (Tree Star, Ashland, OR, USA).

**ELISA assay.** The cells were stimulated with an immobilized anti-TCR-β mAb (3 μl/ml) for 16 h. The concentrations of cytokines were determined using commercial ELISA kits (IL-6; BioLegend, IL-10 and osteopontin; R&D Systems) in accordance with the manufacturer's instructions. The concentration of IFN-γ in the supernatants was determined as described previously[69].

**Quantitative reverse transcriptase polymerase chain reaction (qRT-PCR).** Total RNA was isolated using TRIzol regent, and complementary DNA (cDNA) was synthesized using the superscript VILO cDNA synthesis kit (cat#11754; Thermo Fisher Scientific). qRT-PCR was performed using the Step One Plus Real-Time PCR System (Thermo Fisher Scientific).

**Immunoblotting analyses.** The CD8 T cells were lysed directly with sodium dodecyl sulfate (SDS) sample buffer (0.1 M Tris-HCl, 20 % glycerol, 4% SDS, 0.004% bromophenol blue, 50 mM dithiothreitol (DTT)) and sonicated to shear

DNA. The lysates were separated on an SDS polyacrylamide gel and then subjected to immunoblotting with specific antibodies.

**Incorporation of 2-NBDG.** The CD8 T cells were pulsed with 50 μM 2-NBDG (cat#N13195; Thermo Fisher Scientific) in no-glucose media (cat#185-02864; Wako Chemicals) for 30 min at 37 °C, and the incorporation of 2-NBDG into the cells was assessed by FACS.

**Metabolic profiling.** Metabolome measurements and data processing were performed through a facility service at Human Metabolome Technology Inc. (Yamagata, Japan). Briefly, naive CD8 T cells were stimulated with plate-bound anti-TCR-β mAb plus anti-CD28 mAb in the presence of IL-2 for 36 h. The cells ($5 \times 10^6$ cells) were washed with 5% (w/w) mannitol and then lysed with 800 μl of methanol and 500 μl of Milli-Q water containing internal standards (H3304-1002, Human Metabolome Technology Inc.) and left to rest for another 30 s. The extract was obtained and centrifuged at $2300 \times g$ at 4 °C for 5 min, and then 800 μl of the upper aqueous layer was centrifugally filtered through a Millipore 5-kDa cutoff filter at $9100 \times g$ at 4 °C for 120 min to remove proteins. The filtrate was centrifugally concentrated and re-suspended in 50 μl of Milli-Q water for the capillary electrophoresis-mass spectrometry (CE-MS). Cationic compounds were measured in the positive mode of capillary electrophoresis-time of flight-mass spectrometry (CE-TOFMS) and anionic compounds were measured in the positive and negative modes of capillary electrophoresis-tandem mass spectrometry (CE-MS/MS) in accordance with the methods developed by Soga et al. In some experiments, the glutamate concentration was determined using a Glutamate Colorimetric Assay Kit (cat#K629-100; BioVision, Milpitas, CA, USA).

The extracellular acidification rate (ECAR) and oxygen consumption rate (OCR) were measured using an Extracellular Flux Analyzer XFp (Agilent Technologies, Santa Clara, CA, USA). The culture medium was changed to Seahorse XF RPMI Base Medium (Cat#103336-100) before the analysis. Activated CD8 T cells ($1 \times 10^5$) were adhered on Cell Tak coated seahorse 8-well plate and were pre-incubated at 37 °C for 60 min in the absence of $CO_2$. The ECAR were measured at the baseline and in response to 10 mM glucose, 1 μM oligomycin and 50 mM 2-DG (XFp glycolysis stress test kit; cat#103017-100; Agilent Technologies). The OCR was measured under basal conditions and in response to 1 μM oligomycin, 2 μM FCCP, and 0.5 μM rotenone/antimycin A (XFp mito stress test kit; cat#103010-100; Agilent Technologies).

**Detection of senescence-associated β-galactosidase activity.** CD8 T cells were cultured for 12 days as described above, and then an SA β-galactosidase assay was performed using a Senescence β-Galactosidase Staining Kit (cat#9860; Cell Signaling Technology).

**Adoptive transfer of CD8 T cells and *Listeria* infection.** To assess the phenotype of in vivo activated antigen-specific CD8 T cells, naive CD8 T cells were prepared from the spleens of WT OT-1 Tg mice (Thy1.2$^+$) or *menin* KO OT-1 Tg mice (Thy1.2$^+$) and intravenously transferred into naive C57BL/6 (Thy1.1$^+$) mice ($1 \times 10^5$ cells per mouse). The mice were then infected with *Listeria monocytogenes*-expressing OVA (*Lm*-OVA) strain at $5 \times 10^3$ colony-forming units (CFU). The donor cells in the spleen were gated, and the phenotype was assessed on day 7 after *Lm*-OVA.

In the assessment of the secondary immune response of CD8 T cells, naive CD 8 T cells were purified from the spleen of OT-I Tg Thy1.1$^+$ or Thy1.2$^+$ mice and were stimulated with anti-TCR-β mAb plus anti-CD28 mAb in the presence of IL-2 with or without rapamycin (10 nM) for 2 days before being further expanded with IL-2 in the absence of rapamycin for an additional 3 days in vitro. The expanded cells were then mixed at a 1:1 ratio (Thy1.1$^+$:Thy1.2$^+$) and intravenously transferred ($1 \times 10^6$ cells per mouse) into double-congenic (Thy1.1$^+$/Thy1.2$^+$) mice. Twenty days after the transfer, the recipient mice were infected with the *Lm*-OVA strain at $5 \times 10^3$ CFU. The donor cells were prepared and analyzed on day 5 after infection. In several experiments, the donor cells were isolated from the spleen of recipient mice on day 35, mixed at a 1:1 ratio (Thy1.1$^+$:Thy1.2$^+$), and intravenously transferred ($1 \times 10^5$ cells per mouse) into double-congenic (Thy1.1$^+$/Thy1.2$^+$) mice. The following day, the recipient mice were infected with the *Lm*-OVA strain at $5 \times 10^3$ CFU. The donor cells were prepared and analyzed on day 5 after infection.

**RNA sequencing.** The total RNA input material was used to make sequence libraries using the Illumina TrueSeq RNA Sample preparation kit v2 (Illumina, San Diego, CA, USA). Sequencing was performed on an Illumina MiSeq paired-end at 75 bp using the MiSeq Regent kit v3 (Illumina). Reads were aligned to mm10 using TopHat and assembled into transcripts using Cufflinks.

**Primers and probes for qRT-PCR.** The primer and TaqMan probes used for the detection of *Gapdh* were purchased from Thermo Fisher Scientific. The specific primers and Roche Universal Probes were as follows: *Alox5* 5′-GTCAAAATCAGC AACACTATATCTGAG-3′ (forward), 5′-GGAACTGGTAGCCAAACATGA-3′ (reverse), probe #4, *Cd3ε* 5′-CCAGCCTCAAATAAAAACACG-3′ (forward), 5′-G ATGATTATGGCTACTGCTGTCA-3′ (reverse), probe #10, *Ctsg* 5′-ACGGTTCT

GGAAAGATGCAG-3′ (forward), 5′-TCTCGGCCTCCAATGATCT-3′ (reverse), probe #15, *Got1* 5′-GCTGTGCTTCTCGCCTAGTT-3′ (forward), 5′-AAGACTGC ACCCCTCCAAC-3′ (reverse), probe #64, *Hk2* 5′-CAACTCCGGATGGGACA G-3′ (forward), 5′-CACACGGAAGTTGGTTCCTC-3′ (reverse), probe #21,, *Ldha* 5′-GGCACTGACGCAGACAAG-3′ (forward), 5′-TGATCACCTCGTAGGCAC TG-3′ (reverse), probe #12, *Mcpt8* 5′-GGATGTTCCTGCTCCTGGT-3′ (forward), 5′-TGGGGTTTGGACTCTGTACC-3′ (reverse), probe #83, *Psat1* 5′-CCGGTGG ATGTTTCCAAGT-3′ (forward), 5′-GGTCATCCCGGACAATCA-3′ (reverse), probe #21, and *Slc2a1* 5′-GACCCTGCACCTCATTGGT-3′ (forward), 5′-GAT GCTCAGATAGGACATCCAAG-3′ (reverse), probe #99

**Statistical analyses.** Student's t-test or Welch's t-test (metabolic profiling) was used for the statistical analyses.

**Data availability.** The RNA sequencing data can be downloaded with the GEO accession GSE90106. All other data presented in this article are available in the main and supplementary figures, or upon request from the authors.

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

## Acknowledgements

We thank A. Tamai and D. Shimizu for their excellent technical assistance. This work was supported by the JSPS KAKENHI Grant Numbers 15K19133, 15K15155, 16K19158, 17H05794, 17H0486, 17K15728, and 18H05036, and the Mochida Memorial Foundation for Medical and Pharmaceutical Research, and the Takeda Science Foundation.

## Author contributions

J.S. performed the experiments, analyzed the data and wrote the manuscript, T.Y., S.M., S.N., M.K., N.T., A.T., S.M., M.K. and M.Y. performed and supported the experiments, K.I. and Y.I. established and provided the *Utx*<sup>flox/flox</sup> mice, and M.Y. conceptualized the research, directed the study and edited the manuscript.

## Additional information

**Competing interests:** The authors declare no competing interests.

