## [Peer Review File · Nature Communications]

Reviewers' comments:

Reviewer #1 (Remarks to the Author):

The study by Suzuki et al. aims to address the role of menin in regulating T cell senescence. For these investigation the authors use T cells from a menin knockout mouse, which they activate in vitro before assessing phenotype and function. Together the data presented in the manuscript argue that menin regulates senescence of CD8 T cells. While the absence of menin indeed alters activation, function and proliferation of CD8 T cells, the emphasis on menin regulating senescence appears to be misplaced in the text, as menin seems to most definitively alter mTOR activation and subsequent glutamine metabolism. The data are clear and concise and support the conclusion that menin deficiency changes how CD8 T cells respond to stimulation. However, the mechanism of this direct interaction between mTOR and menin is not sufficiently explored. This lack of direct mechanistic evidence in conjunction with some problematic experimental design lead us to recommend major revisions specifically outlined below.

Major Concerns:

- 1) The authors begin the introduction with a much-appreciated description of the slowly acquired age-related senescence they are referring to throughout the manuscript. However, it is not clear that the rapid and robust stimulation used in this in vitro system is an appropriate model for this type of senescence. Indeed, these cells appear to be more sensitive to stimulation in vitro, resulting in more divisions at lower doses of stimulation, rapid production of effector molecules etc. It would seem to be appropriate to evaluate these CD8 T cells in response to a lower stimulating dose.
- 2) The authors have access to LM-OVA, is there a reason they use in vitro stimulation rather than in vivo stimulation prior to evaluating phenotype and cytokine production at day 7?
- 3) Are the kinetics of the menin KO and WT responses simply different? The menin KO cells become activated, proliferate and produce cytokines faster. Is the data in Figure 3, which is used to support the claim that the KO cells become pre-maturely senescent, in fact a result of a concomitant faster transition from effector phase to contraction?
- 4) Figure 3F: The authors have already stated that KO cells established fewer memory CD8 T cells than WT. In the current experiment you must show how many cells were present at D0 of the secondary stimulation (the infection). A better experiment to test memory expansion and function would be to infect mice with LM-OVA, at memory (D30+) transfer an equal number of WT and KO cells from primary infected mice into different naïve mice and infect with LM-OVA. While not a perfect secondary infection, at least in this model the memory cells would be starting at equal numbers.
- 5) Figure 4 evaluates different metabolites are in vitro stimulation, is it possible to perform this after in vivo stimulation? While these differences occur in vitro, it might be altered by the availability of glutamine in vivo. Is the amount of glutamine used in vitro comparable to a physiological concentration in serum or in a lymph node?
- 6) How does menin, a tumor suppressor scaffold protein regulate mTOR? Is this regulation found in contexts of immunological senescence in vivo such as the inflammatory environment of diet induced obese mice?

Minor Concerns:

- 1) Summary currently says, "The increase supplementation of a-ketoglutarate..." This phrasing makes it very unclear as to whether you are referring to exogenously provided a-ketoglutarate or an endogenous increase in a-ketoglutarate as a result of increased glutamine metabolism. In general, the phrasing of such interactions is difficult to decipher throughout the manuscript.
- 2) The discussion could be improved by adding an explanation of how the authors perceive this to function in physiological scenarios of senescence for example in the elderly. Is menin reduced in older T cells, resulting in reduced glutamine uptake and increased senescence

Reviewer #2 (Remarks to the Author):

Suzuki et al. describe that deficiency of the tumor suppressor menin in CD8 T cells accelerates effector cell differentiation through hyperstimulating the AKT-mTORC pathway and inducing a hypermetabolic state including the glutamine \rightarrow α -ketoglutarate. Increased α -ketoglutarate influences H3K27 demethylation through utx-1. Overall, data quality is high and the data support the proposed model. The link to senescence is less convincing, mainly due to the use of this term. The authors define senescence as the expression of various cytokines and the increase of SA-beta-gal activity. Expression of inflammatory cytokines in T cells is conceptually different from senescence-associated secretory phenotype described for fibroblasts undergoing cellular senescence. Moreover, the increased SA-beta-gal activity was not dependent on the epigenetic effects. In fact, the authors may observe T cell exhaustion rather than senescence. The authors should therefore soften their conclusions on "senescence" in the title, abstract and introduction and should include a discussion that the term "senescence" is rather ambiguous for T cells. Few points should be addressed.

1. The data appear to be mostly explained to be downstream of the increased known AKT activity in menin-deficient cells. This should be better reflected in title and abstract.
2. Figure 4b describes increased Glu after 24 hours in menin KO mice, conversely, Glu in Fig. 4d in KO mice is decreased after 36 hours. Is the difference between the two figures explained by rapid kinetic changes?
3. Although Figure 1 does not show a difference in frequencies of IFNG-producing cells, Figure 6 shows that IFNG MFI is increased suggesting that IFNG is included in the inflammatory phenotype of menin-deficient T cells.
4. The discussion on direct menin effects on BACH2 expression is very soft, in particular at the end of the discussion, and not supported by data. Moreover, decreased BACH2 levels, if they can be confirmed, may be due to increased AKT activity that induces BACH2 phosphorylation and degradation.

RESPONSES TO THE REVIEWERS

Reviewer #1

(Remarks to the Author)

The study by Suzuki et al. aims to address the role of menin in regulating T cell senescence. For these investigation the authors use T cells from a menin knockout mouse, which they activate in vitro before assessing phenotype and function. Together the data presented in the manuscript argue that menin regulates senescence of CD8 T cells. While the absence of menin indeed alters activation, function and proliferation of CD8 T cells, the emphasis on menin regulating senescence appears to be misplaced in the text, as menin seems to most definitively alter mTOR activation and subsequent glutamine metabolism. The data are clear and concise and support the conclusion that menin deficiency changes how CD8 T cells respond to stimulation. However, the mechanism of this direct interaction between mTOR and menin is not sufficiently explored. This lack of direct mechanistic evidence in conjunction with some problematic experimental design lead us to recommend major revisions specifically outlined below.

Response:

We appreciate this reviewer's critical reading and suggestions. Based on the suggestions below, we have performed several new experiments and changed the title to, "The tumor suppressor menin prevents effector CD8 T cell dysfunction via metabolic restriction by targeting mTOR activation."

Major Concerns:

1) The authors begin the introduction with a much-appreciated description of the slowly acquired age-related senescence they are referring to throughout the manuscript. However, it is not clear that the rapid and robust stimulation used in this in vitro system is an appropriate model for this type of senescence. Indeed, these cells appear to be more sensitive to stimulation in vitro, resulting in more divisions at lower doses of

stimulation, rapid production of effector molecules etc. It would seem to be appropriate to evaluate these CD8 T cells in response to a lower stimulating dose.

Response: As suggested, we evaluated the phenotype of *menin*-deficient CD8 T cells in response to a lower stimulating dose of anti-TCR- β mAb (**new Supplementary Fig. S3**). A decreased CD62L, CD27 and CD226 expression and increased PD-1 level were detected in *menin*-deficient CD8 T cells with low-dose anti-TCR- β /anti-CD28 mAb stimulation compared with control WT CD8 T cells. The production of osteopontin was also increased in *menin* KO CD8 T cells with low-dose anti-TCR- β mAb stimulation.

2) The authors have access to LM-OVA, is there a reason they use in vitro stimulation rather than in vivo stimulation prior to evaluating phenotype and cytokine production at day 7?

Response: We previously reported that the frequency and absolute number of OVA-specific CD8 T cells were significantly lower in the spleen of *Lm*-OVA-infected *menin*-deficient mice than in *Lm*-OVA-infected WT mice (Yamada et al. *J. Immunol.* 197: 4079, 2016). In the present manuscript, we reported that a reduced expression of CD27 was detected five days after *Lm*-OVA infection. However, we were unable to examine the cytokine production, as the number of OVA-specific CD8 T cells was limited *in vivo*. Therefore, we evaluated the cell surface phenotype and cytokine production in *Lm*-OVA-infected *menin*-deficient CD8 T cells at day 7 (**new Fig. 1g, 1h, 1i and new Supplementary Fig. S4**). A decreased CD62L and CD27 expression and increased PD-1 level were detected in CD8 T cells from *Lm*-OVA-infected

menin-deficient mice. The expression of OPN was increased in CD8 T cells from *Lm*-OVA-infected T cell-specific *menin*-deficient mice compared with WT mice, whereas the IFN- γ expression was comparable between two strains. In addition, we assessed the change in the surface phenotype of *menin*-deficient CD8 T cells over time after TCR stimulation *in vitro* (**new Supplementary Fig. S2**).

3) *Are the kinetics of the menin KO and WT responses simply different? The menin KO cells become activated, proliferate and produce cytokines faster. Is the data in Figure 3, which is used to support the claim that the KO cells become pre-maturely senescent, in fact a result of a concomitant faster transition from effector phase to contraction?*

Response: As suggested, we observed the change in the surface phenotype of *menin*-deficient CD8 T cells over time after TCR stimulation *in vitro* (**new Supplementary Fig. S2**). The expression of CD62L, CD27, PD-1 and CD226 in *menin*-deficient naïve CD8 T cells was comparable that in WT naïve CD8 T cells. The senescent phenotype was detected in *menin*-deficient CD8 T cells as early as day3 after the initial stimulation, whereas the senescence-like phenotype was not observed in WT CD8 T cells until at least day 12. We believe that these results indicate a low possibility of a concomitant faster transition from the effector phase to contraction in *menin* KO effector CD8 T cells.

4) *Figure 3F: The authors have already stated that KO cells established fewer memory CD8 T cells than WT. In the current experiment you must show how many cells were present at D0 of the secondary stimulation (the infection). A better experiment to test memory expansion and function would be to infect mice with LM-OVA, at memory*

(D30+) transfer an equal number of WT and KO cells from primary infected mice into different naïve mice and infect with LM-OVA. While not a perfect secondary infection, at least in this model the memory cells would be starting at equal numbers.

Response: As suggested, we transferred an equal number of WT and *menin*-deficient memory CD8 T cells from primary infected mice into different naïve mice. The recipient mice were then infected with *Lm*-OVA, and the number of OVA-specific CD8 T cells was assessed (**new Fig. 3g** and **Supplementary Fig. S9**). The expansion of *menin*-deficient memory CD8 T cells was significantly reduced compared with that of WT memory CD8 T cells. Furthermore, the *in vitro* treatment of *menin* KO CD8 T cells with rapamycin during the initial TCR-mediated activation phase restored secondary expansion of OVA-specific *menin*-deficient CD8 T cells upon *Lm*-OVA infection. These results show that the activation status of mTORC1 during primary TCR-stimulation determines the expansion of memory CD8 T cells upon receiving secondary TCR stimulation.

5) Figure 4 evaluates different metabolites are in vitro stimulation, is it possible to perform this after in vivo stimulation? While these differences occur in vitro, it might be altered by the availability of glutamine in vivo. Is the amount of glutamine used in vitro comparable to a physiological concentration in serum or in a lymph node?

Response: Because of the technical limitation, we were unable to measure intracellular glutamine level in *in vivo* stimulated CD8 T cells. Instead, we measured the glutamine level in the serum (**Figures for reviewers 1a**). The concentration of glutamine in serum

was approximately 1 mM, while that in the complete RPMI1640 medium was 3 mM. We performed the experiments under 1 mM conditions *in vitro* and obtained a similar senescent phenotype in *menin*-deficient CD8 T cells (**Figures for reviewers 1b, 1c and 1d**).

6) How does menin, a tumor suppressor scaffold protein regulate mTOR? Is this regulation found in contexts of immunological senescence in vivo such as the inflammatory environment of diet induced obese mice?

Response: We showed that the activity of Akt and mTOR was increased in *menin* KO activated CD8 T cells during the initial activation phase compared with WT CD8 T cells. The treatment with an Akt inhibitor showed only a limited inhibitory effect on the induction of a senescence-like phenotype in *menin* KO CD8 T cells (**Figures for reviewers 2**), whereas the senescent phenotype was almost completely suppressed by rapamycin (**new Fig. 3**). The α -KG-dependent activation of mTORC1 has been reported in other cell lines (Duran et al. *Oncogene* 32:4549, 2013). We confirmed that α -KG induces the activation of the mTORC signaling in activated CD8 T cells (**new Fig. 7a and 7b**). Menin likely regulates mTOR via two pathways (**new Fig. 7g**). First, menin regulates mTOR phosphorylation by targeting Akt activation during the TCR-mediated initial activation phase, as was previously demonstrated in non-immune cells. The enhanced mTORC1 activation accelerates the cellular metabolism, and potentially resulting in the upregulation of the intracellular concentration of α -KG. The increased intracellular α -KG supports the prolonged activation of mTOR signaling and the

subsequent central carbon metabolism. Thus, α -KG and mTOR form a positive feedback loop to induce histone H3K27 demethylation and subsequent dysfunction in *menin* KO CD8 T cells. We have now touched on this point in the DISCUSSION section in the revised manuscript.

At present, whether or not the *menin*-mediated regulation of mTOR is involved in the *in vivo* immunological senescence, such as the establishment of an inflammatory environment of diet-induced obese mice, is unclear. However, we confirmed the increased phosphorylation of S6 protein, an indicator of mTORC1 activity, in *menin*-deficient CD8 T cells from *Lm*-OVA-infected mice (**new supplementary Fig. 6b**). These results clearly show the enhanced mTORC1 activity in *menin*-deficient CD8 T cells after antigen recognition *in vivo*.

Minor Concerns:

1) Summary currently says, “The increase supplementation of *a*-ketoglutarate...” This phrasing makes it very unclear as to whether you are referring to exogenously provided *a*-ketoglutarate or an endogenous increase in *a*-ketoglutarate as a result of increased glutamine metabolism. In general, the phrasing of such interactions is difficult to decipher throughout the manuscript.

Response: As suggested, we weakened the expression concerning the involvement of α -KG in the premature cellular senescence (dysfunction) in *menin* KO CD8 T cells throughout the revised manuscript. We also changed the title to, “The tumor suppressor *menin* prevents effector CD8 T cell dysfunction via metabolic restriction by targeting mTOR activation.”

2) The discussion could be improved by adding an explanation of how the authors perceive this to function in physiological scenarios of senescence for example in the elderly. Is menin reduced in older T cells, resulting in reduced glutamine uptake and increased senescence.

Response: As suggested, we included this point in the DISCUSSION section of the revised manuscript. In addition, we added the data on the reduction of cytosolic menin protein in older effector CD8 T cells (**new supplementary Fig. 5**).

Reviewer #2

(Remarks to the Author)

Suzuki et al. describe that deficiency of the tumor suppressor menin in CD8 T cells accelerates effector cell differentiation through hyperstimulating the AKT-mTORC pathway and inducing a hypermetabolic state including the glutamine – alpha-ketoglutarate. Increased alpha-ketoglutarate influences H3K27 demethylation through utx-1. Overall, data quality is high and the data support the proposed model. The link to senescence is less convincing, mainly due to the use of this term. The authors define senescence as the expression of various cytokines and the increase of SA-beta-gal activity. Expression of inflammatory cytokines in T cells is conceptually different from senescence-associated secretory phenotype described for fibroblasts undergoing cellular senescence. Moreover, the increased SA-beta-gal activity was not dependent on the epigenetic effects. In fact, the authors may observe T cell exhaustion rather than senescence. The authors should therefore soften their conclusions on “senescence” in the title, abstract and introduction and should include a discussion that the term “senescence” is rather ambiguous for T cells.

Response:

We appreciate this reviewer’s helpful comments, which enabled us to improve the quality of our manuscript.

As was pointed out, the definition of T cell senescence is not fully established. The unresponsiveness of T cells is subdivided into several statuses, which are characterized by terms such as anergy, exhaustion and senescence. The phenotype of *menin* KO CD8 T cells is most similar to that reported for human senescent T cells, since the down-regulation of CD28 and irreversible growth arrest are observed. However, we

revised the term “senescence” to “senescence-like phenotype” or “dysfunction” in the revised manuscript, as suggested.

Few points should be addressed.

1. The data appear to be mostly explained to be downstream of the increased known AKT activity in menin-deficient cells. This should be better reflected in title and abstract.

Response: As was pointed out, the activity of Akt and mTOR was increased in *menin*-deficient activated CD8 T cells during the initial activation phase compared to WT CD8 T cells. However, treatment with an Akt inhibitor showed only a limited inhibitory effect on the induction of dysfunction in *menin* KO CD8 T cells (**Figures for reviewers 2**), whereas the dysfunction was almost completely suppressed by rapamycin treatment during the initial TCR-mediated activation phase (**new Fig. 3**). Furthermore, we found that α -KG induced the prolonged activation of mTOR signaling and the central carbon metabolism in activated CD8 T cells (**new Fig. 7**). Based on these findings, we suspect that menin regulates the mTOR signaling via two pathways (**new Fig. 7g**). First, menin regulates mTOR phosphorylation by targeting Akt activation during the TCR-mediated initial activation phase. The enhanced mTORC1 activation accelerates glutamine metabolism, potentially resulting in the upregulation of the intracellular concentration of α -KG. The increased intracellular α -KG supports the prolonged activation of mTOR signaling and the subsequent central carbon metabolism. Thus, α -KG and mTOR form positive feedback loop to induce histone H3K27

demethylation and subsequent cellular senescence (dysfunction) in *menin*-deficient CD8 T cells. We have now touched on this point in the DISCUSSION section in the revised manuscript.

For this reason, we changed the title to, “The tumor suppressor *menin* prevents CD8 T cell dysfunction via metabolic restriction by targeting mTOR activation.”

2. Figure 4b describes increased Glu after 24 hours in menin KO mice, conversely, Glu in Fig. 4d in KO mice is decreased after 36 hours. Is the difference between the two figures explained by rapid kinetic changes?

Response: As was pointed out, the intracellular glutamate level was increased in *menin*-deficient CD8 T cells compared with WT cells at 24 h after the initial stimulation. In contrast, the level of glutamate was low in *menin*-deficient CD8 T cells at 36 h. To clarify this point, we measured the time-dependent changes in the intracellular concentration of glutamate and glutamine (**Figures for reviewers 3**). The intracellular glutamate concentration of WT CD8 T cells, increased over time after the TCR stimulation while the glutamate level in *menin* KO CD8 T cells peaked at 24 h and decreased by 48 h. The level of glutamate was higher in *menin* KO CD8 T cells than in WT cells at 24 h. However, the intracellular glutamate level was lower in *menin* KO CD8 T cells than in WT CD8 T cells at 48 h. The glutamine level peaked at 24 h after TCR stimulation in both WT and *menin* KO CD8 T cells. Similar to that of glutamate, the intracellular glutamine level was higher in *menin* KO CD8 T cells than in WT cells, but lower at 48 h. Glutamine serves as a source of carbon and nitrogen for the synthesis

of proteins, nucleotides, lipids and amino acids in proliferating cells, in part via glutaminolysis. Naïve CD8 T cells do not divide within 24 h after the initial TCR stimulation, and then cells divide at least twice from 24 to 48 h. We demonstrated that *menin*-deficient CD8 T cells divided more quickly and robustly responded to TCR stimulation than control WT cells (**new Fig. 2d**). Therefore, it is likely that the intracellular glutamate and glutamine levels were decreased in *menin*-deficient activated CD8 T cells that robustly proliferated within 36 h after TCR stimulation.

3. Although Figure 1 does not show a difference in frequencies of IFNG-producing cells, Figure 6 shows that IFNG MFI is increased suggesting that IFNG is included in the inflammatory phenotype of *menin*-deficient T cells.

Response: As was pointed out, a marginal increase in IFN- γ production in *menin*-deficient CD8 T cells was detected by an ELISA (**new Fig. 6e**). However, the IFN- γ production was 20% higher in *menin*-deficient CD8 T cells than in WT CD8 T cells. We also previously reported the increased IFN- γ production in *menin*-deficient CD4 T cells (Kuwahara et al. *Nat. Commun.* 5: 3555, 2014). The augmentation of IFN- γ production by menin deficiency was more severe in CD4 T cells (approximately 200% increase) than in CD8 T cells (approximately 20% increase). In addition, the generation of IFN- γ -producing cells *in vivo* induced by *Lm*-OVA infection in T cell-specific *menin*-deficient mice was comparable to that in WT mice (**new Fig. 1i**). We therefore excluded the ELISA data of IFN- γ production from the *menin*-deficient CD8 T cells. A menin deficiency may preferentially enhance the IFN- γ production from CD4 T cells

through the Menin-Bach2 axis, which may be involved in the immune dysfunction observed in T cell-specific *menin* KO mice.

4. The discussion on direct menin effects on BACH2 expression is very soft, in particular at the end of the discussion, and not supported by data. Moreover, decreased BACH2 levels, if they can be confirmed, may be due to increased AKT activity that induces BACH2 phosphorylation and degradation.

Response: As was pointed out, our hypothesis regarding of the role of bach2 in the induction of senescence-like dysfunction in *menin*-deficient CD8 T cells was not supported by the experimental data. We therefore deleted this paragraph from the DISCUSSION section in the revised manuscript.

We assessed the effect of an Akt inhibitor on Bach2 protein expression in *menin*-deficient CD8 T cells (**Figures for reviewers 4**). Although the Bach2 level in WT activated CD8 T cells was moderately increased by the treatment with an Akt inhibitor, this Akt inhibitor failed to increase the Bach2 protein level in both the cytosolic and nuclear fractions of *menin* KO CD8 T cells. These results suggest that augmented Akt signaling is not involved in the reduced Bach2 expression in *menin* KO activated CD8 T cells.

Reviewers' comments:

Reviewer #1 (Remarks to the Author):

The authors have satisfied my concerns.

****Comments on Reviewer#2 report****

I agree with reviewer 2 comment:

'The revisions do not provide evidence that alpha-ketoglutarate activates mTORC1 in a positive feedback loop; it is equally possible that fuel deprivation leads to metabolic stress and AMPK activation that is bypassed with alpha-ketoglutarate.'

The authors can test this possibility by assessing mtorc1 activity using AMPK deficient cells or using respiratory chain inhibitors plus alpha ketoglutarate.

Reviewer #2 (Remarks to the Author):

The revisions by Suzuki et al. have only incompletely addressed my questions.

The use of the term senescence-like remains equivocal. How are these cells different from terminal effector T cells?

The metabolic results can mostly be explained as increased stimulation and mTORC1 activity. The title is therefore not very instructive.

The claim that alpha-ketoglutarate directly regulates mTORC activity is not supported. Results could be explained by energy depletion and activation of AMPK.

As eluded to by Reviewer 1, the cells could be hyperreactive and proliferate faster. As shown for the Glu concentrations, kinetic differences could be important. Of note, it is not mentioned when some of the metabolic studies were done and how they relate to turnover. The data that menin inhibits mTORC1 activation and that menin-deficient cells are hyperstimulated and differentiate into effector T cells is convincing, but not entirely novel and extend their previous data of an influence on the balance of terminal effector T cells versus memory precursor cells in an immune response. The observation that mTORC1 inhibition by rapamycin favors memory precursor cells goes back to studies by Rafi Ahmed and others. Whether the cells are truly senescent or just terminal effector T cells remains semantic and is not easily solved by calling them senescent-like. The metabolite results are likely to just reflect increased metabolic activity with increased glycolysis and TCA activity that is dependent on glutamine as a fuel. The revisions do not provide evidence that alpha-ketoglutarate activates mTORC1 in a positive feedback loop; it is equally possible that fuel deprivation leads to metabolic stress and AMPK activation that is bypassed with alpha-ketoglutarate.

The manuscript remains unfocused, the observation of α -ketoglutarate influencing H3K27 demethylation is of interest but unclear how directly related to menin and the discussion on BACH2 and SIRT6 is not very helpful.

Several issues on data quality were not addressed in the cover letter, and I did not check whether they were resolved.

RESPONSE TO REVIEWER 1

(Remarks to the Author)

The authors have satisfied my concerns.

Response:

We appreciate this reviewer's critical reading and judgment.

(Comments on Reviewer#2 report)

I agree with reviewer 2 comment:

'The revisions do not provide evidence that alpha-ketoglutarate activates mTORC1 in a positive feedback loop; it is equally possible that fuel deprivation leads to metabolic stress and AMPK activation that is bypassed with alpha-ketoglutarate.'

The authors can test this possibility by assessing mtorc1 activity using AMPK deficient cells or using respiratory chain inhibitors plus alpha ketoglutarate.

Response:

First, we examined whether or not the deprivation of glutamine during the TCR-mediated activation phase induces AMPK activation. We determined the AMPK activation by the phosphorylation status of AMPK α . As shown in **new supplementary Fig. 14**, the deprivation of glutamine did not induce AMPK phosphorylation (AMPK α [Thr172] and AMPK α 1 [Ser485]/AMPK α 2 [Ser491]) but did reduce the phosphorylation level. The level of AMPK α protein was also decreased by the deprivation of glutamine and restored by α -KG administration. These results indicate that the activation status of an AMPK is not connected with the glutamine- α -KG-mediated regulation of mTOR phosphorylation. Consequently, we

concluded that the glutamine- α -KG axis controls the mTOR-signaling through an AMPK-independent pathway.

We also assessed the effect of α -KG on mTOR phosphorylation in activated CD8 T cells under glutamine-deprived conditions (36 h after initial stimulation) in the presence of the AMPK inhibitor, dorsomorphin. Unexpectedly, the treatment of activated CD8 T cells under glutamine-deprived conditions with an AMPK inhibitor for 2 h induced a striking reduction in mTOR protein. The reduction of mTOR by AMPK treatment was not observed in CD8 T cells cultured under glutamine-sufficient control conditions (**Figure for reviewers 1**). Furthermore, treatment with A76966 (30 μ M), an AMPK activator, failed to suppress the induction of dysfunction in *menin*-deficient CD8 T cells (**Figure for reviewers 2**). These results support the notion that the deprivation of glutamine inhibits the dysfunction of *menin*-deficient CD8 T cells without inducing AMPK activation.

In addition, as suggested, we treated activated CD8 T cells with respiratory chain inhibitors under glutamine-deprived conditions in the presence or absence of α -KG. However, we could not assess mTORC1 activity, since the treatment of CD8 T cells with respiratory chain inhibitors under glutamine-deprived conditions rapidly induced cell death.

RESPONSE TO REVIEWER 2

(Remarks to the Author)

The revisions by Suzuki et al. have only incompletely addressed my questions.

The use of the term senescence-like remains equivocal. How are these cells different from terminal effector T cells?

The metabolic results can mostly be explained as increased stimulation and mTORC1 activity. The title is therefore not very instructive.

The claim that alpha-ketoglutarate directly regulates mTORC activity is not supported. Results could be explained by energy depletion and activation of AMPK.

As eluded to by Reviewer 1, the cells could be hyperreactive and proliferate faster. As shown for the Glu concentrations, kinetic differences could be important. Of note, it is not mentioned when some of the metabolic studies were done and how they relate to turnover.

The data that menin inhibits mTORC1 activation and that menin-deficient cells are hyperstimulated and differentiate into effector T cells is convincing, but not entirely novel and extend their previous data of an influence on the balance of terminal effector T cells versus memory precursor cells in an immune response. The observation that mTORC1 inhibition by rapamycin favors memory precursor cells goes back to studies by Rafi Ahmed and others. Whether the cells are truly senescent or just terminal effector T cells remains semantic and is not easily solved by calling them senescent-like. The metabolite results are likely to just reflect increased metabolic activity with increased glycolysis and TCA activity that is dependent on glutamine as a fuel. The revisions do not provide evidence that alpha-ketoglutarate activates mTORC1 in a positive feedback loop; it is equally possible that fuel deprivation leads to metabolic stress and AMPK activation that is bypassed with alpha-ketoglutarate.

The manuscript remains unfocused, the observation of α -ketoglutarate influencing H3K27 demethylation is of interest but unclear how directly related to menin and the discussion on BACH2 and SIRT6 is not very helpful.

Several issues on data quality were not addressed in the cover letter, and I did not check whether they were resolved.

Response:

As suggested, we have changed the title to, “The tumor suppressor menin prevents effector CD8 T cell dysfunction by targeting mTORC1-dependent metabolic activation.” In addition, we deleted the word “senescent” and “senescent-like” in the RESULTS and DISCUSSION section of the revised manuscript, since the definition of T cell senescence has not been fully established.

We performed several experiments to assess the involvement of AMPK activation in the effects of glutamine-deprivation and α -KG addition. First, we examined whether or not the deprivation of glutamine during the TCR-mediated activation phase induces AMPK activation. We determined the AMPK activation by the phosphorylation status of AMPK α . As shown in **new supplementary Fig. 14**, the deprivation of glutamine did not induce AMPK phosphorylation (AMPK α (Thr172) and AMPK α 1 [Ser485]/AMPK α 2 [Ser491]) but did reduce the phosphorylation level. The level of AMPK α protein was also decreased by the deprivation of glutamine and restored by α -KG administration. These results indicate that the activation status of an AMPK is not connected with the glutamine- α -KG-mediated regulation of mTOR phosphorylation. Consequently, we concluded that the glutamine- α -KG axis controls the mTOR-signaling through an AMPK-independent pathway.

We also assessed the effect of α -KG on mTOR phosphorylation in activated CD8 T cells under glutamine-deprived conditions (36 h after initial stimulation) in the presence of the AMPK inhibitor dorsomorphin. Unexpectedly, the treatment of activated CD8 T cells under glutamine-deprived conditions with an AMPK inhibitor for 2 h

induced a striking reduction in mTOR protein. The reduction of mTOR by AMPK treatment was not observed in CD8 T cells cultured under control glutamine-sufficient conditions (**Figure for reviewers 1**). Furthermore, treatment with A76966 (30 μ M), an AMPK activator, failed to suppress the induction of dysfunction in *menin*-deficient CD8 T cells (**Figure for reviewers 2**). These results support the notion that the deprivation of glutamine inhibits dysfunction of *menin*-deficient CD8 T cells without inducing AMPK activation.

We felt that the phenotype of *menin* KO CD8 T cells was most similar to that reported for human senescent T cells, given our observations of the the down regulation of CD28, SASP-like gene expression, the induction of SA β -Gal activity and irreversible growth arrest. However, as pointed out, the definition of T cell senescence has not been fully established, and the difference between senescent CD8 T cells and terminally differentiated CD8 T cells remains unclear. Therefore, we revised the term “senescence” and “senescence-like phenotype” to “dysfunction” in the RESULTS and DISCUSSION sections of the revised manuscript.

As was pointed, our manuscript do not provide direct evidence that α -KG activates mTORC1 in a positive feedback loop. Therefore, we modified the text regarding the α -KG-mediated activation of mTORC1 in the ABSTRACT, RESULTS and DISCUSSION sections of the revised manuscript. Furthermore, we mentioned our

findings that rapamycin inhibits the α -KG-dependent induction of the ECAR and OCR (New Fig. 7f).

In addition, we discussed the molecular mechanism by which α -KG activates the mTORC1-signaling in the second paragraph of the DISCUSSION section of the revised manuscript. In brief, we discussed the possibility that α -KG activates mTORC1-signaling in CD8 T cells through enzymatic conversion to the 2-hydroxyglutarate (2-HG), a metabolite of α -KG, given that the 2-HG dependent activation of mTOR-signaling has been reported. We found that the concentration of 2-HG was significantly higher in *menin*-deficient activated CD8 T cells than in WT CD8 T cells (Figure for reviewers 3a), and the 2-HG level was reduced by glutamine-deprivation and restored by α -KG supplementation in WT activated CD8 T cells (Figure for reviewers 3b). The accumulation of R- and S-2-HG in CD8 T cells after receiving TCR stimulation has also been reported, and the production of 2-HG in antigen-stimulated CD8 T cells is dependent on extracellular glutamine. Furthermore, it was reported that 2-HG is produced by lactate dehydrogenase A (*Ldha*) and/or malate dehydrogenase. We showed that the expression of *Ldha* was reduced by glutamine deprivation and induced in an α -KG-dependent manner in activated CD8 T cells (new Fig. 7e). We also showed that the ECAR was increased by α -KG in CD8 T cells cultured under glutamine-deprived conditions, indicating the augmented enzymatic activity of *Ldha* by α -KG (new Fig. 7c). Both the α -KG-induced expression of *Ldha* (new Fig. 7e) and the augmentation of the ECAR (new Fig. 7f) were inhibited by

rapamycin. These findings suggest that α -KG activates mTORC1-signaling in CD8 T cells through enzymatic conversion to 2-HG.

We depleted glutamine from the culture medium only during initial activation phase (approximately 72 h). Under these conditions, the dysfunction of *menin*-deficient CD8 T cells was almost completely restored (**new Fig. 5**). Furthermore, the deprivation of glutamine during initial activation phase augmented histone H3K27 methylation, and the effect of glutamine deprivation was antagonized by α -KG. These results indicate that glutamine was not only consumed as a fuel but also used as a source of epigenetic regulator. Our experimental results showed that the di- and tri-methylation levels were reduced, and glutamine metabolism was augmented in *menin* KO CD8 T cells. Furthermore, the dysfunction in *menin*-deficient CD8 T cells was partially restored by deletion of the *utx* gene. These results suggest that accelerated H3K27 me2/me3 demethylation is involved in the dysfunction observed in the *menin*-deficient activated CD8 T cells. Thus, the prolonged activation of the mTORC1 signal in *menin* KO CD8 T cells seems to accelerate the histone H3K27me2/me3 demethylation via the augmentation of glutamine metabolism, thereby inducing dysfunction. We feel that this is an important point for understanding the molecular mechanism of dysfunction in *menin*-deficient activated CD8 T cells. Therefore, we have now mentioned our findings regarding α -KG influencing H3K27 demethylation and *menin/utx* double-deficient mice in the manuscript.

As suggested, we have now deleted the discussion on SIRT6 from the revised manuscript and moved the NADH data to a Supplementary figure (**new supplementary Fig. 15c**). In addition, we modified the discussion on H3K27 methylation and deleted the discussion on Bach2 in the revised manuscript.

REVIEWERS' COMMENTS:

Reviewer #2 (Remarks to the Author):

The authors have done an excellent job to address the questions that I have raised.

RESPONSE TO REVIEWER 2

(Remarks to the Author)

The authors have done an excellent job to address the questions that I have raised.

Response:

We appreciate this reviewer's critical reading and judgment.